# Indirect climate impacts of the Hunga eruption

Ewa M Bednarz[1,2], Amy H. Butler[2], Xinyue Wang[3], Zhihong Zhuo[4], Wandi Yu[5], Georgiy Stenchikov[6], Matthew Toohey[7], Yunqian Zhu[1,2]

1. Cooperative Institute for Research in Environmental Sciences (CIRES), University of Colorado Boulder, Boulder, CO, USA.
2. NOAA Chemical Sciences Laboratory (NOAA CSL), Boulder, CO, USA.
3. Department of Atmospheric and Oceanic Sciences, University of Colorado Boulder, Boulder, USA
4. Department of Earth and Atmospheric Sciences, University of Quebec in Montreal, Montreal (Quebec), Canada.
5. Lawrence Livermore National Laboratory, CA, USA
6. Physical Science and Engineering Division, King Abdullah University of Science and Technology, Jeddah, Saudi Arabia
7. Institute of Space and Atmospheric Studies, University of Saskatchewan, Saskatoon, Canada.

*Correspondence to*: Ewa M. Bednarz (ewa.bednarz@noaa.gov)

**Abstract.**

Injection of sulfur and water vapour by the Hunga volcanic eruption significantly altered chemical composition and radiative budget of the stratosphere. Yet, whether the eruption could also affect surface climate, especially via indirect pathways, remains poorly understood. Here we investigate these effects using large ensembles of simulations with the CESM2(WACCM6) Earth system model, incorporating interactive chemistry and aerosols in both coupled ocean and atmosphere-only configurations.

We find some statistically significant extratropical regional climate responses to the eruption driven by circulation changes; these are partially linked to the modulation of El Nino Southern Oscillation, and its associated teleconnections, and to perturbations of the stratospheric polar vortex in both hemispheres. The stratospheric anomalies affect surface climate through modulating the North Atlantic Oscillation in the Northern Hemisphere (up to three boreal winters following the eruption) and the Southern Annular Mode in the Southern Hemisphere in late 2023. The latter is partly related to a concurrent reduction in Antarctic ozone, as increased stratospheric aerosols and water vapor reach the polar vortex.

Our study suggests that the eruption could have had a non-negligible influence on regional surface climate, and discusses the mechanisms via which such an influence could occur. However, the results also highlight that this forcing is relatively weak compared to interannual variability, and is subject to model uncertainties in the representation of key processes. More research is thus needed before definitive statements on the role of the eruption in contributing to surface climate and weather events in the following years are made.

## 1. Introduction

The 2022 Hunga volcano erupted on January 15, 2022, injecting large amounts of water vapor and other volcanic materials into the stratosphere and upper atmosphere up to 57 km (Proud et al., 2022). Microwave Limb Sounder remote sensing measurements suggest that the eruption increased global stratospheric water vapor burden by approximately 10-15% (Khaykin et al., 2022), equivalent to ~ 150 Tg of water vapor (Millan et al., 2022), making it the largest water vapor perturbation in the satellite era. At the same time, the eruption injected a moderate amount of SO2, ~0.5-1.0 Tg of $SO_2$ (Carn

et al., 2023; Sellitto et al., 2024), and likely other species, such as sea salt, chlorine (Zhu et al., 2023), and bromine species (Li et al., 2023). The $SO_2$ converted to sulfate aerosol and resulted in the largest stratospheric aerosol optical depth (sAOD) since the 1990 Mt. Pinatubo eruption (Taha et al., 2022), partly due to the presence of anomalous water vapor enhancing aerosol growth in the initial months following the eruption (Zhu et al., 2022; Quaglia et al., 2025).

The large volcanic water vapor and sulfur injections by the eruption were shown to significantly impact stratospheric temperatures and chemistry. Upper stratospheric cooling of up to a few degrees K were observed in 2022 and 2023 as the result of the radiative cooling by the Hunga water (Stocker, et al., 2024; Wang et al. 2023; Randel et al., 2024). The enhanced heterogeneous processing on sulfate aerosols in turn contributed to the strong stratospheric ozone depletion observed at Southern Hemisphere mid-latitudes in the austral winter of 2022 (Zhang et al., 2024). The state-of the art climate

models participating in the Hunga Tonga-Hunga Ha'apai Volcano Impact Model Observation Comparison (HTHH-MOC) Project reproduce the stratospheric temperature and ozone responses as observed when the Hunga forcings are included (Zhuo et al., 2025).

Unlike the significant impacts in the stratosphere, the direct radiative impact of the eruption on the surface climate is likely

small (e.g. Schoerberl et al. 2024; Gupta et al., 2025; Quaglia et al., 2025). However, whether the eruption could lead to regional surface climate changes via other, more indirect pathways remains not well understood. For instance, using WACCM4 climate model simulations, Jucker et al. (2024) reported the existence of robust regional surface climate responses to the stratospheric water injection that emerged 3-8 years following the eruption, albeit without a clear mechanistic understanding of the origin of such responses. With the Hunga water stratospheric burden e-folding time of ~31-

43 months estimated from the current chemistry-climate models (Zhuo et al. 2025), the reasons behind such a significant delay in the emergence of the response are also not well understood.

Here we address this using large 30-member ensembles of CESM2(WACCM6) earth system model simulations with interactive chemistry and aerosols forced with and without Hunga $SO_2$ and $H_2O$ injections; the simulations are also carried

out using either the atmosphere-only or coupled-ocean configuration to investigate the role of atmosphere-ocean coupling. Section 2 gives details on the model used, experimental protocol and presents the resulting evolution of stratospheric

aerosols and water vapour from the eruption. Section 3 discusses the simulated changes in near-surface air temperatures in the simulations. Sections 4 and 5 present changes in the Northern Hemisphere (NH) and Southern Hemisphere (SH) stratospheric polar vortex and their links with the troposphere and the extratropical surface climate. Section 6 summarizes the main results.

## 2. Methods

### 2.1. CESM2(WACCM6) model

We use the Community Earth System Model version 2 coupled to the high-top Whole Atmosphere Community Climate Model version 6 (CESM2(WACCM6); Danabasolglu et al., 2020; Gettleman et al. 2019). The configuration includes an interactive Troposphere-Stratosphere-Mesosphere-Lower-Thermosphere (TSMLT) chemistry scheme (Davis et al., 2023), and the interactive Modal Aerosol Microphysics version 4 (MAM4, Lu et al., 2016). The horizontal resolution of the atmospheric model is 0.9° in latitude by 1.25° in longitude, with 70 vertical levels with a hybrid-pressure coordinate up to ~140 km. The ocean component is the Parallel Ocean Program version 2 (POP2; Danabasoglu et al., 2012; Smith et al., 2010), with 60 vertical levels, and the horizontal resolution is 1.125° in the zonal direction and between about 0.27° and 0.64° in the meridional direction.

### 2.2. Experimental description.

The simulations follow the 'Experiment 1' experimental protocol of the Hunga Tonga-Hunga Ha'apai impacts Model Observation Comparison project (HTHH-MOC) described in detail in Zhu et al. (2025). Briefly, the simulations cover the 10 years following the eruption - i.e. January 2022 to December 2031 - and follow the Coupled Model Intercomparison Project phase 6 (CMIP6) middle-of-the-road SSP2-4.5 emission scenario (Meinshausen et al., 2020). All simulations are initialized from the same initial atmospheric and ocean state using the observed sea-surface temperatures following the procedure described in Richter et al. (2022). The simulations are run with a free-running meteorology, with the exception of the first 1-2 months, depending on the ensemble member (see below), where the atmospheric meteorology is nudged to the Goddard Earth Observing System (GEOS) Modern-Era Retrospective Analysis for Research and Applications, Version 2 (MERRA–2) reanalysis (Gelaro, et al., 2017) in order to ensure the eruption occurs during realistic background conditions.

For each experiment, there are two pairs of simulations: a 'forced' simulation with simultaneous injection of 0.5 Tg $SO_2$ at 20-28 km (with 71% of injection at 20-22 km and 29% of injection at 22-28 km) and 150 Tg $H_2O$ at 25-35 km (with 69% at 25-27 km, 28% at 27-30 km and 5% of 30-35 km) at 22ºS-14ºS and 182ºE-186ºE on 15 January and a 'control' simulation without the Hunga injection. Such altitude profile of the injections has been chosen based on the results of Zhu et al. (2022), where it was found to produce a relatively good agreement with observed distribution of Hunga water and aerosol, albeit with the $SO_2$ injection scaled here to give the total of 0.5 Tg $SO_2$, as recommended by the HTHH-MOC protocol (Zhu et al.

2025). The first pair, denoted here as "HUNGA_fix", is the core atmosphere-only simulation of the HTHH-MOC that uses prescribed climatological sea-surface temperatures and sea-ice (same for both control and perturbation simulation) derived from the NOAA high-resolution blended analysis (Banzon et al., 2022). The second pair, denoted here as "HUNGA_cpl", is analogous but runs in a fully coupled mode with interactive ocean and sea-ice. In order to thoroughly account for and explore the role of interannual variability, each simulation consists of 30 ensemble members - a significant improvement upon the original 10 required by the HTHH-MOC - obtained by modifying the end date of the initial nudging in the increment of one day (with resulting end date between 27 January - 23 February 2022 depending on the member).

We note that initialization of the model with the observed conditions leads to some model drift in the first few years as the model moves away from the imposed observed conditions towards its own quasi-equilibrium state; this is particularly the case for the ocean component of the coupled runs (not shown). However, since exactly the same initial conditions are used in both the control and perturbed experiment, and the Hunga response is always taken as the difference between perturbed and control (rather than looking at absolute values), this to a first order removes any impacts such drift may have on the inferred results.

## 2.3 The simulated evolution of sulfate aerosols and water vapour

The resulting evolution of the anomalous (forced minus control) stratospheric Aerosol Optical Depth (sAOD) and stratospheric water vapour during the first 5 years simulated in the two pairs of experiments is shown in Fig. 1. In general, the coupled ocean and atmosphere-only simulations produce a similar evolution of aerosols and $H_2O$. In 2022, most of the anomalous aerosols and water vapour is confined to the tropics and the SH mid-latitudes. While some aerosols reach also the SH high latitudes in the second part of 2022, most of the Antarctic sAOD and $H_2O$ enhancement does not occur until 2023, when the largest Antarctic sAOD and $H_2O$ anomalies are found. Similarly, the transport of Hunga water and aerosols to the NH does not occur until early 2023. Unlike the aerosols, the anomalous $H_2O$ is uplifted into the ascending branch of the Brewer Dobson Circulation and reaches the upper stratosphere, with peak values at the stratopause simulated in the second part of 2023, and persisting there until ~2027. The evolution of aerosols and water vapor simulated in CESM2(WACCM) is thus similar to that simulated by other models participating in the HTHH-MOC and inferred from available satellite data (Zhuo et al., 2025). The resulting changes in lower, middle and upper stratospheric temperatures are shown in Fig. S1.

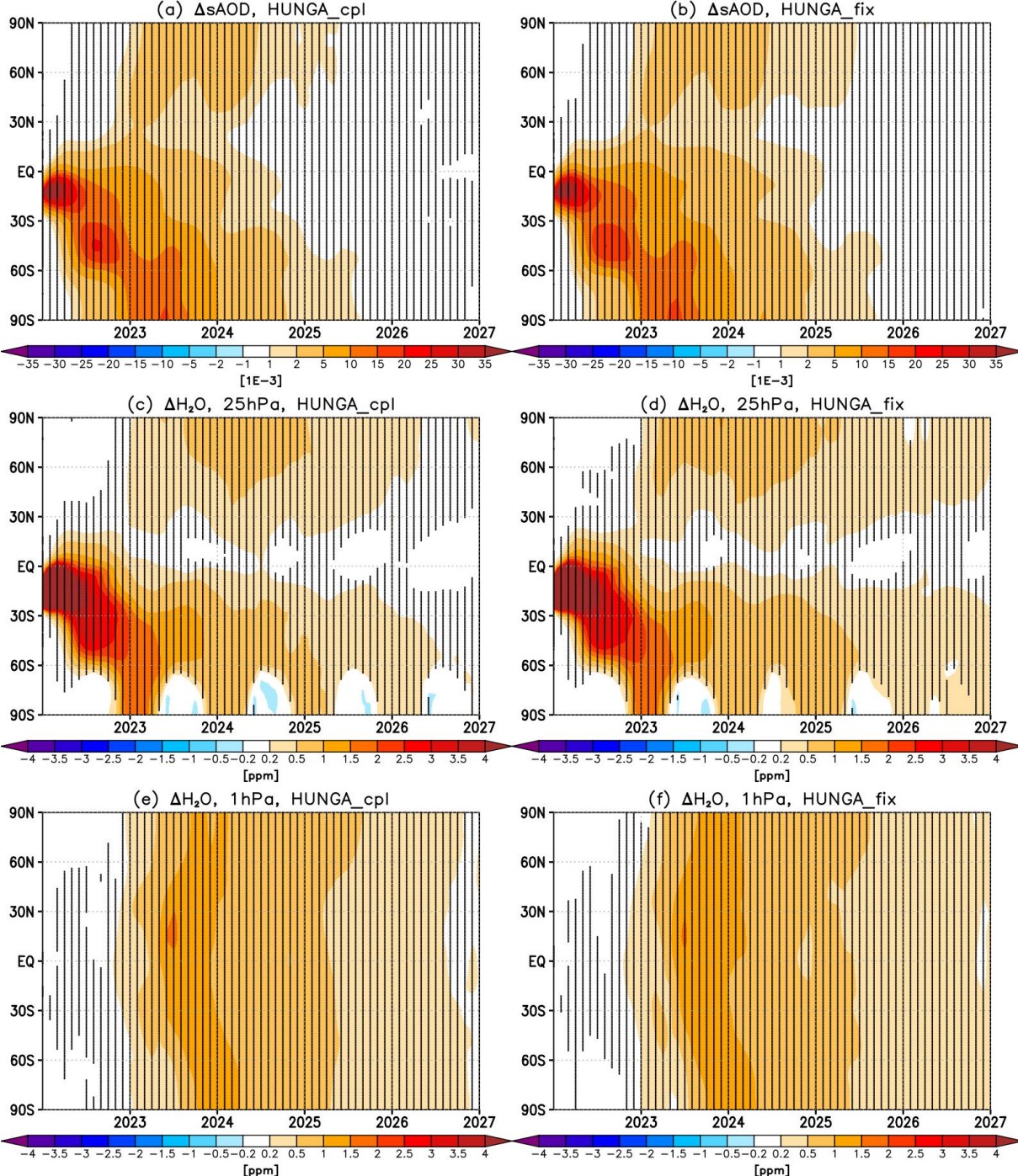

**Figure 1. Evolution of the ensemble mean changes in zonal mean (a-b) sAOD, (c-d) H₂O at 25 hPa and (e-f) H₂O at 1 hPa between the forced simulation and the control. Left panels are for the coupled ocean simulations, and right panels for the atmosphere-only simulations. Stippling denotes regions where the response is statistically significant, here taken as larger than +/- 2 standard errors in the difference in means (≈ 95% confidence level).**

125

## 3. Simulated changes in near-surface air temperatures following the eruption

Figures 2 and 3 show the simulated changes in annual mean surface temperatures for each of the five years following the eruption. The mean over the years 1-3 (2022-2024), i.e. when most of the direct radiative forcing from the eruption is present (e.g. Quaglia et al 2025), is also included. In agreement with the small radiative forcing estimated for the eruption ( -0.27 W/m$^2$ clear-sky top-of-the-atmosphere perturbation averaged between 0-60ºS and 2022-2023 in both simulations, see Quaglia et al., 2025), the simulations do not show any significant cooling in the SH. This is unlike the small SH cooling (~0.1 K) estimated from a radiative transfer and energy balance models in Gupta et al. (2025). However, the coupled ocean simulations (Fig. 2) show some significant and mostly negative temperature anomalies in the NH extratropics, especially over North America, Europe and central Asia, in years 1-3. These NH anomalies are unlikely to be due to the direct radiative forcing from aerosols or water vapor because most of the perturbation is found in the SH (Quaglia et al., 2025); rather, they are likely indicative of atmospheric circulation changes, as will be discussed in more detail in Section 4. In contrast, the atmosphere-only simulations (Fig. 3) show few significant near surface temperature changes. This occurs partly because the use of atmosphere-only set-up constrains and dampens any surface responses; in addition, the associated NH circulation responses are much weaker, as will be discussed in Section 4.

In addition to the extratropical anomalies, the coupled ocean simulations show a significant cooling in the equatorial Pacific in 2022 and 2023; the anomaly dominates the concurrent tropical temperature changes (Fig. 4a,c) and is evident also in the global mean (Fig. 4c). This Pacific response corresponds to a negative phase of the El Nino-Southern Oscillation (ENSO; Trenberth, 1997), or a La Nina-like response. Figure 4(e-f) shows the changes in the Equatorial Southern Oscillation Index (eqSOI), defined as the difference in 5°S-5°N sea-level pressure between 80°W-130°W and 90°E-140°E. Positive changes in eqSOI correspond to La Nina-like responses, and negative changes in eqSOI correspond to El Nino-like responses. Comparison between panels 4e and 4c shows that changes in the eqSOI track and mirror changes in tropical and global mean near-surface temperatures, illustrating that the modulation of ENSO by the eruption is the dominant contributor to the large-scale temperature changes following the eruption.

Our results suggest that the eruption could have contributed to the anomalous persistence of the La Nina-like conditions observed between 2021-2023 (e.g. Iwakiri et. al., 2023). The simulated La Nina-like response in 2022-2023 following the eruption is consistent with previous studies that found a similar response to past SH volcanic eruptions (Pausata et al. 2020, 2023; Ward et al. 2021) and SH wildfire aerosols (Fasullo et al. 2023). These studies concluded that changes in the inter-tropical convergence zone (ITCZ) following a hemispherically asymmetrical aerosol forcing are the main driver of the La Nina-like response, and it is thus likely that the same mechanism operates here. Note that the ENSO response (and hence tropical temperature response) changes sign in year 3 and 4 (2024-2025), with a significant El Nino-like response in the Pacific, suggesting that the Hunga forcing might have longer-term impacts due to the much longer timescales governing

ocean processes, with the subsequent development into El Nino-like response likely arising due to delayed negative ocean-atmosphere feedbacks that terminate the mature La Nina phase (e.g.Wang, 2001, 2018). We note that the simulated El Nino-like response appears around a year later than the real-world phase shift to El Nino in 2023-2024.

Shifts in the ENSO phase are associated with shifts in tropical convective heating that drive poleward-propagating planetary waves (Hoskins and Karoly, 1981), forming teleconnections that remotely affect regional climate patterns around the world (Mo and Livezey, 1986). In the observations, La Nina is associated with anomalous high pressure over the North Pacific and southeastern US, low pressure over Canada, and a dipole in pressure over the North Atlantic that arises in part due to ENSO's stratospheric pathway (Butler et al. 2014, Domeisen et al. 2019). This circulation pattern generally means colder conditions over northwestern North America and subtropical Asia, and warmer conditions over the southeastern US and northern Eurasia during boreal winter. La Nina teleconnections to the SH can similarly lead to cooling over the Maritime continent, southern Africa, and northern South America. These patterns are qualitatively present in the coupled simulation responses to the eruption, especially in 2023 (year 2) and to some extent in the year 1-3 average (Figure 2f), suggesting that these regional responses may arise in part from the shift towards a more La Nina-like teleconnection in the first two years. That these responses are largely not significant may be a reflection of either sampling error – previous studies have noted that some regional ENSO teleconnection responses are subject to large internal variability (Deser et al. 2017, 2018) - or an inability of coupled models to capture ENSO teleconnections adequately (Williams et al. 2023, Fang et al. 2024). Such anomalies are not found in the atmosphere-only simulations since, by definition, ENSO (and its related teleconnections) is the same as in the control experiment (Fig. 3, Fig. 4b,d,f).

Using an earlier version of WACCM in an atmosphere-only configuration and simulating $H_2O$ injection only, Jucker et al. (2024) reported the existence of robust regional surface climate responses to the stratospheric water injection that emerged 3-8 years following the eruption (so years 4-9 using the same numbering convention as in our study). In stark contrast, our atmosphere-only simulations do not show any robust near-surface air temperature perturbations in either years 1-5 (Fig. 3) or 6-10 (Fig. S3) of the simulations. While a small number of localized anomalies that appear statistically significant can be found in certain individual years, these occur sparsely and incoherently (i.e do not extend to more than a single year in a row). As such these are likely just a manifestation of interannual variability rather than indicating the forced response to the eruption; this is especially true in later part of the simulations where most of the anomalous aerosol and water vapor is removed from the stratosphere (see Fig. 1). Broadly similar conclusions are reached if seasonal mean responses are used instead of annual means (not shown).

The role of potential Hunga modulation of the ENSO variability and its teleconnections in contributing to some of the surface climate responses, as found here for the coupled ocean simulations (Fig. 2 and S2), has also been suggested by Jucker et al. (2024) using a second set of experiments with a medium complexity general circulation model MiMa. In their

case, however, the response resembled a positive ENSO – or El Nino-like – response in years 4-9, unlike the La Nina-like response in year 1-2 here followed by El Nino-like response in year 4 (as well as a La Nina-like response again in years 8-9; see Fig. S2). The simulated ENSO changes in later parts of our simulations are likely not indicative of long-term changes in the Hunga forcing itself (which weakens over time) but rather are the long-term result of the initial perturbation and how it influences the oscillatory nature of ocean-atmosphere ENSO feedback in the long-term. Given that ENSO is not perfectly periodic and there are many uncertainties in the details of such long-term modulation due to stochastic noise influencing these different feedbacks at different times (e.g. Wang, 2001), there are many uncertainties in the exact timing of the different responses in the model, and so we focus here mostly on the results in the initial few years following the eruption.

In addition to the ENSO changes in the coupled ocean simulations, which can drive remote extratropical responses in regional surface climate via teleconnections, the anomalous stratospheric aerosols and water vapor could drive changes in stratospheric circulation that then influence extratropical surface climate. In the next two sections, we consider stratospheric polar vortex response to the Hunga forcing in the Northern (section 4) and Southern (section 5) hemisphere.

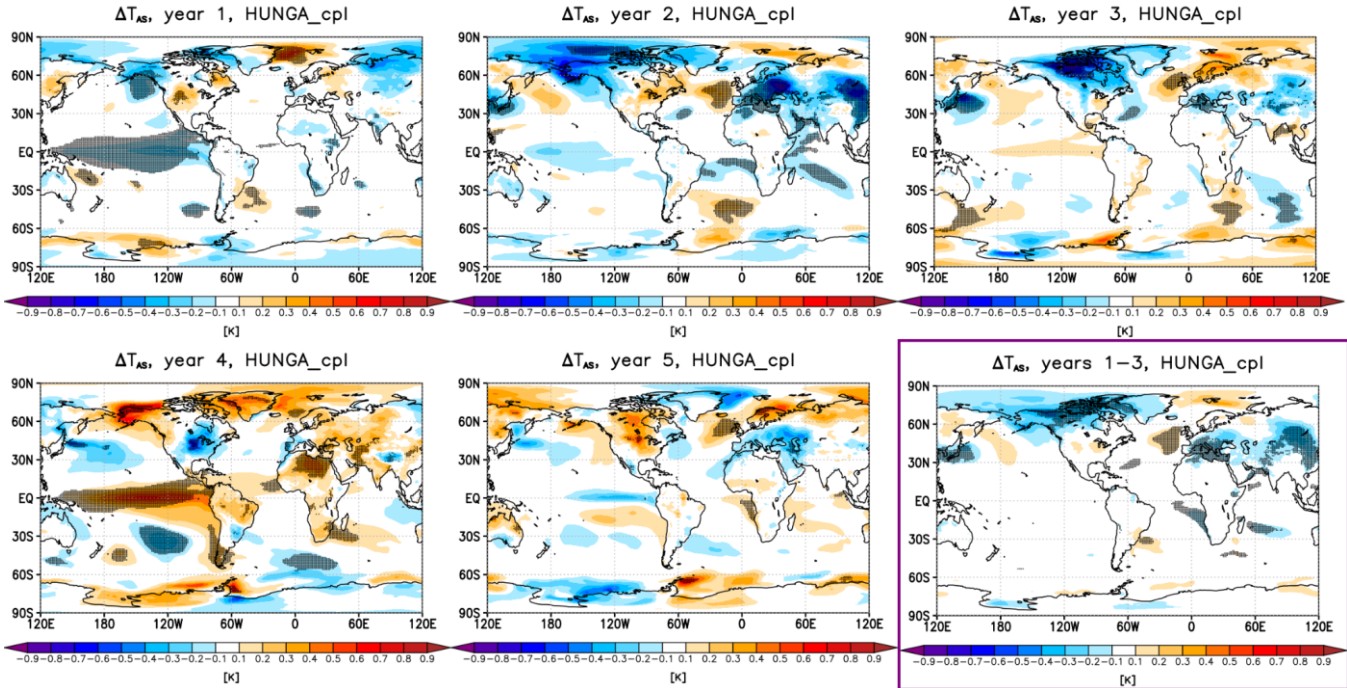

**Figure 2. Yearly mean changes in near-surface air temperature between the forced simulation and the control in the coupled ocean simulations for each of the 5 years following the eruption (i.e. 2022-2026). The bottom right panel shows the response averaged over the first 3 years (i.e. 2022-2024). Stippling indicates statistical significance (defined as in Fig. 1). See Figure S2, supplement, for the corresponding changes in years 6-10.**

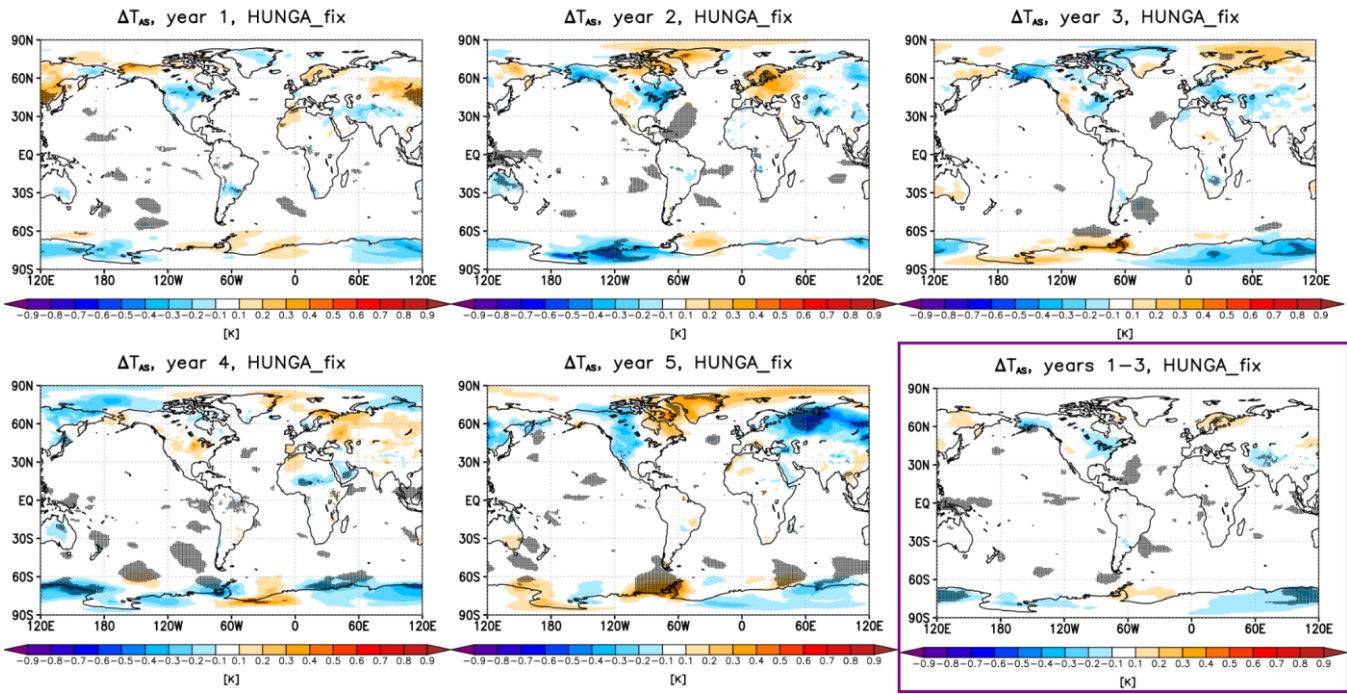

**Figure 3.** As in Fig. 2 but for the changes in the atmosphere-only simulations. See Figure S3, supplement, for the corresponding changes in years 6-10.

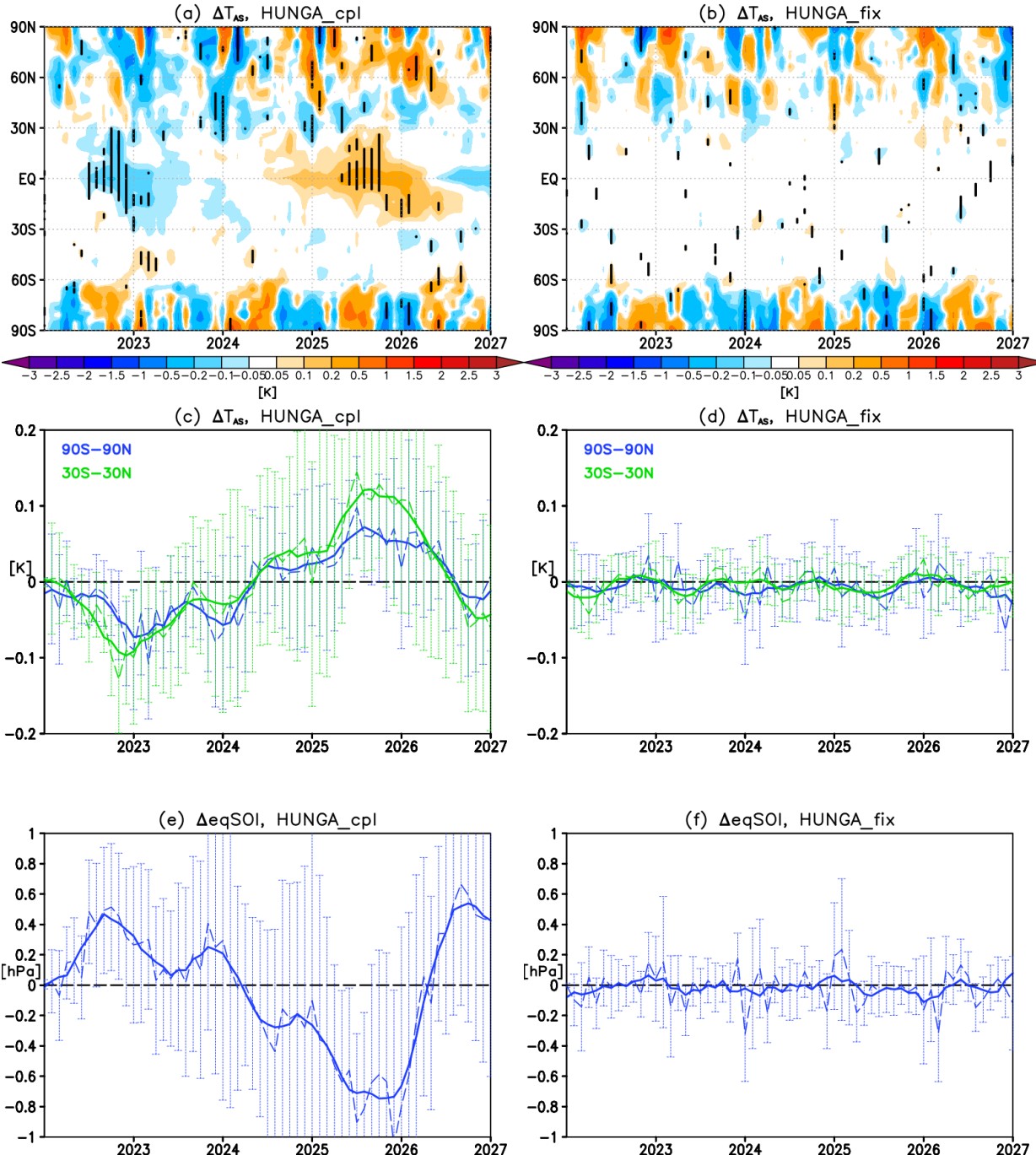

**Figure 4.** Time series of changes in (a-b) zonal mean near-surface air temperature, (c-d) global and tropical mean near-surface air temperature, and (e-f) the eqSOI index between the forced simulation and the control in the coupled ocean (left) and atmosphere-only (right) simulations. Dashed lines in (c-f) indicate ensemble mean changes and solid lines their 5-month running means. Error bars in (c-f) denote confidence intervals of the ensemble mean changes (+/- 2 standard errors).

## 4. Seasonal changes in the NH polar vortex and impacts on the extratropical surface climate

This section examines the response of the NH stratospheric vortex to the eruption and the resulting impacts on the NH extratropical surface climate. Figure 5 shows the evolution of anomalies in zonal winds at 65°N during the 5 years following the eruption, as well as the associated changes in geopotential heights over the Arctic polar cap (65°N-90°N). We find a statistically significant strengthening of the stratospheric vortex in the coupled ocean simulations in late 2022 (Fig. 5a); the response propagates down to the troposphere and is accompanied by a statistically significant reduction in geopotential

heights over the Arctic (so colder, more compact air) at the same time (Fig. 5c). At the surface, the response manifests as the pattern of changes in sea-level pressure projecting onto the positive phase of the Arctic Oscillation (AO) and, for the Atlantic sector, the North Atlantic Oscillation (NAO; Hurrell et al., 2003), i.e. decrease in sea-level pressure over the Arctic and increase over the NH mid-latitudes in early winter (November-December-January, NDJ; Fig. 6a). The response is also accompanied by some statistically significant near-surface air temperature changes, especially a cooling over Greenland

(Fig. 6d). The stratospheric and surface responses change sign in late winter (February-March-April, FMA), with a weakening (albeit not statistically significant in the zonal means) of the zonal winds (Fig. 5a) and a pattern of changes in sea-level pressure projection onto the negative phase of NAO (Fig. 7a). The response is also accompanied by statistically significant surface temperature changes, including a cooling over Europe and warming over Eastern Canada (Fig. 7d).

While the analogous early winter zonal wind changes in the second (i.e. 2023/2024) and third winter (2024/2025) are not statistically significant (Fig. 5a), the pattern of changes in sea-level pressure and near-surface air temperatures also resemble those associated with a positive NAO, i.e. increase in sea-level pressure in the NH mid-latitudes (Fig. 6b-c), a warming over northern Eurasia and a cooling over southern Europe and continental Asia (Fig. 6e-f). In the late period of the third winter, the coupled ocean simulations show a significant weakening of the stratospheric vortex (Fig. 5a) and increase in geopotential

heights over the Arctic (Fig. 5c); the response propagates down to the surface and results in a negative AO/NAO pattern (Fig. 7c) as well as increases in surface temperatures in southern Europe (Fig. 7f).

To summarise, while there is some variability between the individual winters, all three Arctic winters following the eruption simulated in the coupled ocean simulations suggest a strengthening of the stratospheric polar vortex and tropospheric polar

jet stream in early winter, with the accompanied positive NAO-like anomalies in sea-level pressure and near-surface temperatures (with the opposite sign stratospheric and surface climate responses found in late winters). The modulation of the polar vortex by the eruption could be driven by changes in meridional temperature gradients and wind shear associated with anomalous stratospheric radiative heating and cooling from Hunga aerosols and water vapour, and the resulting wave-mean flow feedbacks with planetary wave propagation and breaking. The increased stratospheric aerosols act to increase

tropical lower stratospheric temperatures (Fig. S1a-b) and thus meridional temperature gradients at these altitudes; by thermal wind relationship this can act to strengthen the polar vortex in the lower stratosphere. Evidence for this mechanism

has been found for previous sulfur-rich volcanic eruptions (Polvani et al., 2019; Paik et al., 2023), although some studies point that this polar vortex signal only emerges for eruptions with large aerosol loading (Azoulay et al. 2021; DallaSanta and Polvani, 2022), much larger than the 0.5 Tg $SO_2$ injected in these simulations. However, a unique aspect of the Hunga eruption was the exceptional water vapor injection. The increased stratospheric water vapour results in cooling of the mid-to-upper stratosphere (Fig. S1c-f), with the upper stratospheric cooling being amplified by the $H_2O$-induced ozone reduction at those levels and the resulting reduction in ozone shortwave heating (Randel et al. 2024). This may act to reduce meridional temperature gradients at these altitudes, thereby potentially weakening the upper stratospheric jet. Kuchar et al. (2025) suggested that the initial upper stratospheric anomaly could then propagate down to the lower stratosphere and troposphere and affect NH high latitude surface climate in spring. . The enhancement of stratospheric $H_2O$ also contributes to the warming in the tropical lower stratosphere below the water plume by trapping and re-radiating the outgoing terrestrial radiation, and thus amplifying the aerosol-induced heating there (e.g. Wang et al., 2023; Yook et al., 2025). Nonetheless, full details behind the relative interplay of changes in meridional temperature gradients and wind shear to Hunga water vapour and aerosols, and their coupling with wave feedbacks, on the polar vortex strength remains to be fully understood.

The simulated changes in the polar vortex are also likely to be at least partly related to the modulation of ENSO by the eruption, and the associated changes in tropospheric wave flux to the stratosphere (e.g. Domeisen et al., 2019). In accord, the atmosphere-only simulations (which, by definition, have no ENSO response) show NH polar vortex changes that have different seasonality in winters 1 and 3 to the coupled ocean simulations, with statistically significant strengthening of the stratospheric vortex found mainly in late winter periods (Fig. 5b). This suggests that the modulation of ENSO and its teleconnections plays an important contribution to the NH polar vortex changes simulated after the eruption in the coupled ocean experiments.

In the atmosphere only simulation, where ocean feedbacks cannot interfere with any top-down response, in the second winter the simulations show statistically significant strengthening of the early winter vortex followed by statistically not significant weakening in late winter. Such response might be indicative of aerosol-induced lower stratospheric warming (Fig. S1b) dominating the response in early winter and upper stratospheric cooling (Fig. S1f) dominating the vortex behavior in late winter, with the latter consistent with the mechanism postulated in Kuchar et al. (2025). However, more idealized studies would be needed to diagnose the details of such modulation. In addition, the stratospheric anomalies in the atmosphere-only simulations do not appear to propagate readily to the troposphere, and so any changes in sea-level pressure and surface temperature are much weaker and mostly not statistically significant (Fig. 8; Fig. S4).It is likely that the use of atmosphere-only set up substantially constrains and dampens any surface responses, highlighting that  interpreting the inferred signatures, or their absence, of the Hunga eruption on climate using atmosphere-only model simulations needs to be done carefully. Significant differences in the simulated NH stratospheric vortex responses between the coupled ocean and atmosphere-only simulations were also previously reported in the context of idealized high latitude volcanic eruptions (Guðlaugsdóttir et al., 2025).

The extratropical atmospheric circulation also exhibits substantial variability across timescales, and so we examine the role of interannual variability and ensemble size for the detectability of the climate signals from the eruption (e.g. Maycock and Hitchcock, 2015; Bittner et al., 2016; Milinski et al., 2020 used in different contexts). Figures 9-10 show the range of possible early and late winter NH lower stratospheric wind and NAO changes that can be inferred for the first winter following the eruption if a smaller number of ensemble members than the full ensemble is used. In particular, for each reduced ensemble size, 2000 artificial ensemble means are created by randomly sampling the available 30 members with replacement. The means, +/- 2 standard deviation confidence intervals as well as maximum and minimum values of the resulting 2000 inferred responses are shown in each case.

Inspection of Figure 9 shows that if only 10 ensemble members are used, then in most cases the envelope of potential NH high latitude "ensemble-mean" responses spans both positive and negative values. This highlights the challenge of isolating the relatively weak Hunga impact from the much larger natural interannual variability of the Arctic winter stratosphere, and underscores the need for caution when interpreting surface climate impacts of the eruption inferred both from observations and/or reanalysis (which effectively constitute just one ensemble member) or from model simulations with insufficient ensemble size. For instance, Kuchar et al (2025) used a 10-member ensemble of Hunga simulations with the SOCOLv4 model, and reported a late winter weakening of the NH polar vortex from the eruption, with the associated impacts on the high latitude surface climate. Assuming similar magnitude of both the forced response and natural variability in SOCOLv4 and CESM2(WACCM6), our results suggest that 10 members is not enough to infer the footprint of the eruption on the NH winter vortex with any confidence. In CESM2(WACCM6), the variability is particularly large for the late winter vortex response (i.e. even larger than in early winter; compare left and right panels in Fig. 9), consistent with added uncertainty in the vortex final warming date.

The confidence intervals around the mean response become narrower as more ensemble members are used. Nonetheless, even if close to 30 members are used, the edge of the +/- 2 standard deviation envelope of the distribution tends to be close to 0, and an individual, albeit an outlier, ensemble configuration usually exists (indicated by the max/min lines) that can indicate an opposite response instead. This shows that while interannual variability is likely to have a first order influence on the response inferred from an ensemble of smaller (e.g. 10 members) size, it can likely still have a non-negligible contribution even if as much as 30 members are used. Given that the observational record of the real world response constitutes effectively just one realization, this makes attribution of any regional climate response to the eruption virtually impossible without using a large ensemble.

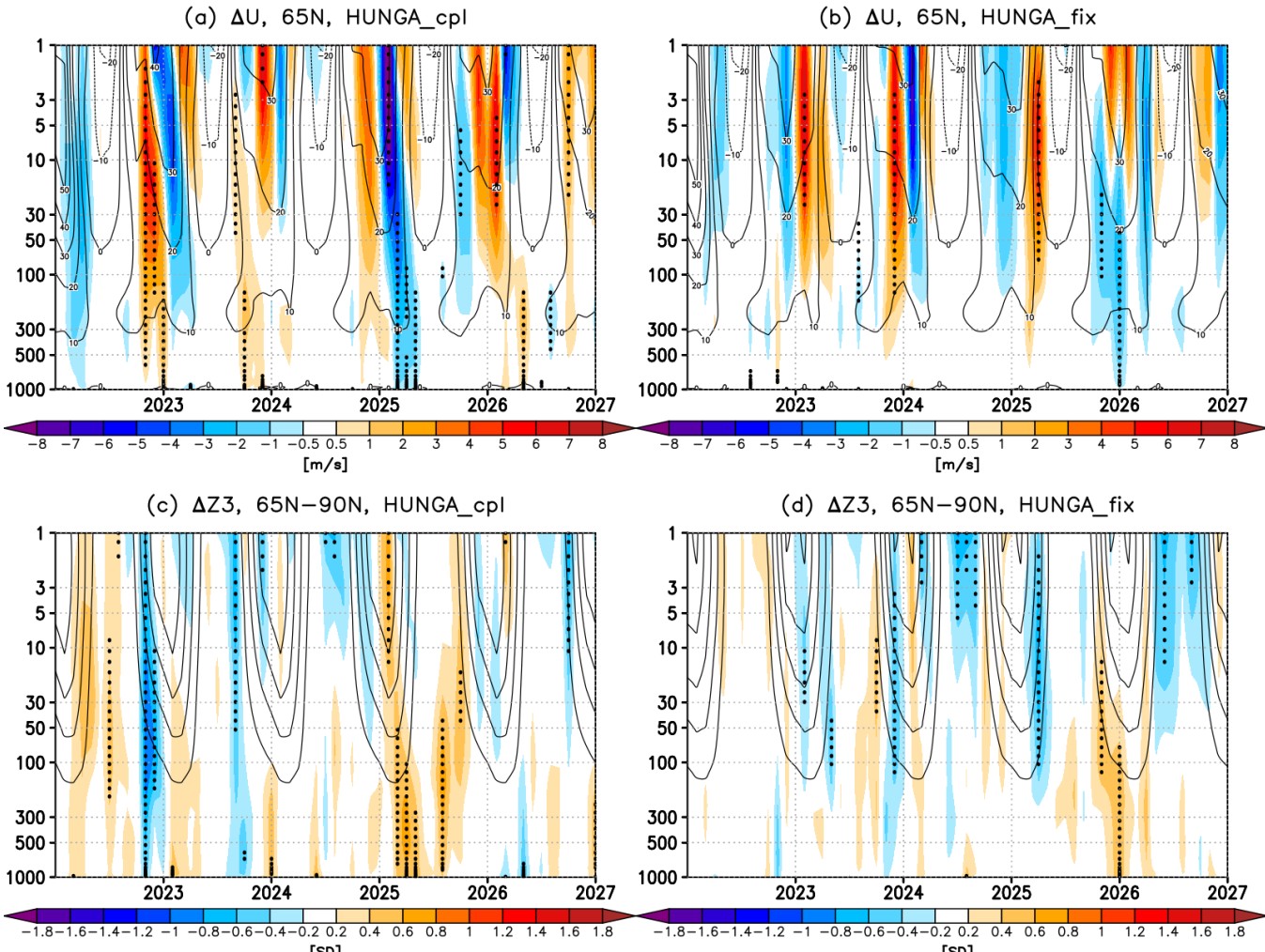

**Figure 5. Evolution of the NH polar vortex. Shading: Time series of ensemble-mean changes in (a-b) zonal wind at 65°N [in units of m/s] and (c-d) geopotential height [in units of standard deviations] averaged over the polar cap (65°N-90°N) between the forced simulation and the control. The left panels are for the coupled ocean simulations and the right panels are for the atmosphere-only simulations. Stippling denotes statistical significance (defined as in Fig. 1). Contours indicate values in the control for reference.**


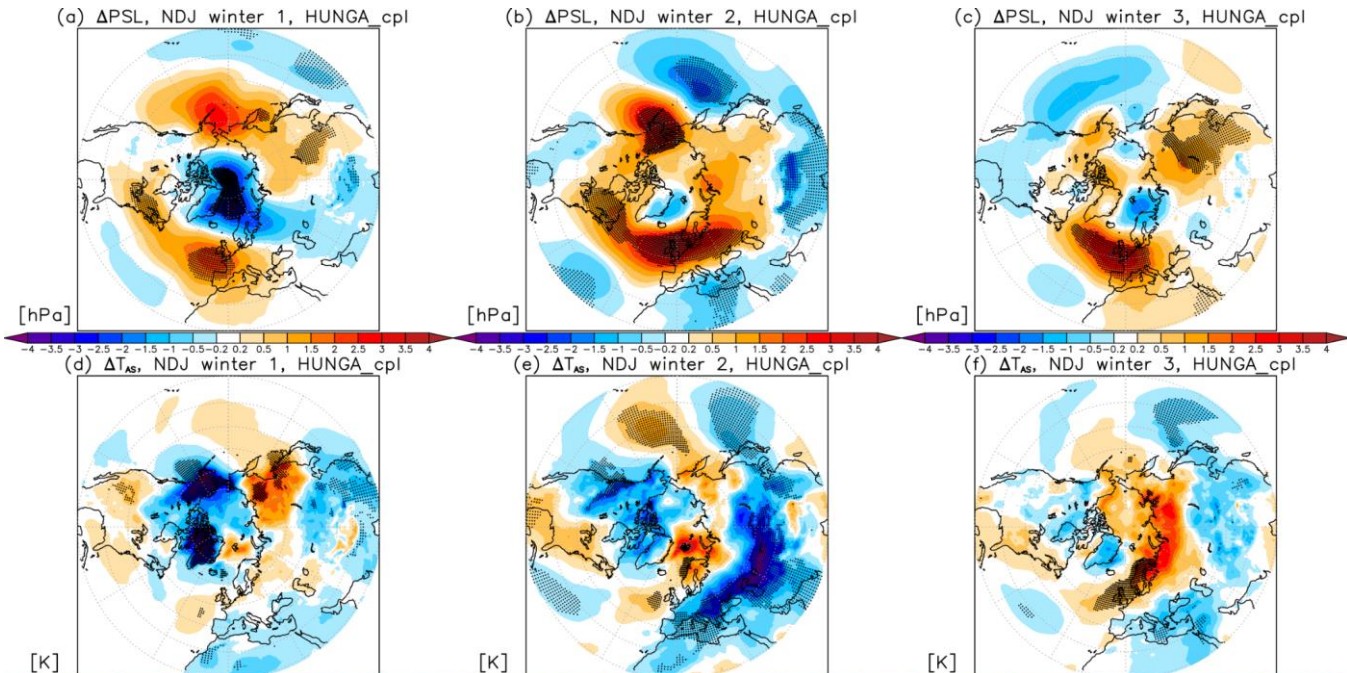

**Figure 6. Changes in the early NH winter (November-December-January) ensemble mean (top) sea-level pressure and (bottom) near-surface air temperature between the forced simulation and the control in the coupled ocean simulations for the first (i.e. 2022/2023), second (i.e. 2023/2024) and third (2024/2025) winter following the eruption (columns). Stippling denotes statistical significance (defined as in Fig. 1).**



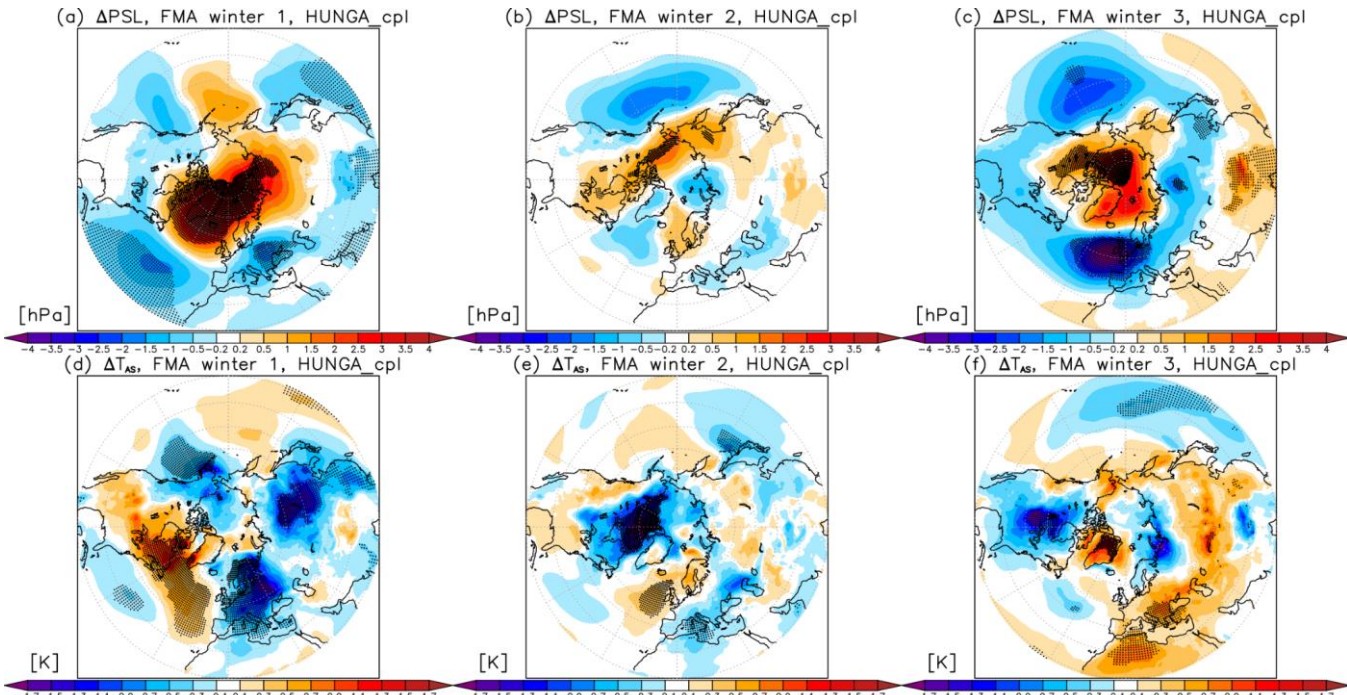

**Figure 7. As in Fig. 6 but for the corresponding late NH winter (February-March-April) changes. See Fig. S4 in the Supplement for the corresponding changes in the atmosphere-only simulations.**


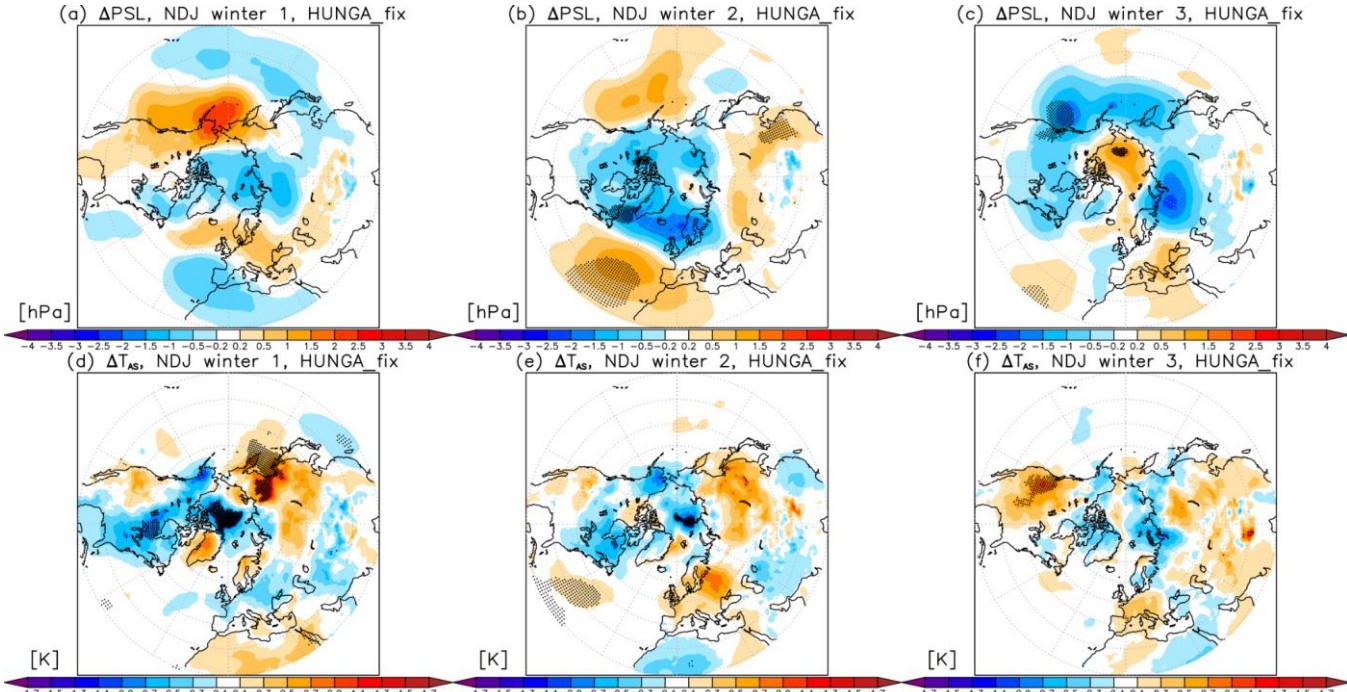

**Figure 8. As in Fig. 6 but for the changes in the atmosphere-only simulations.**


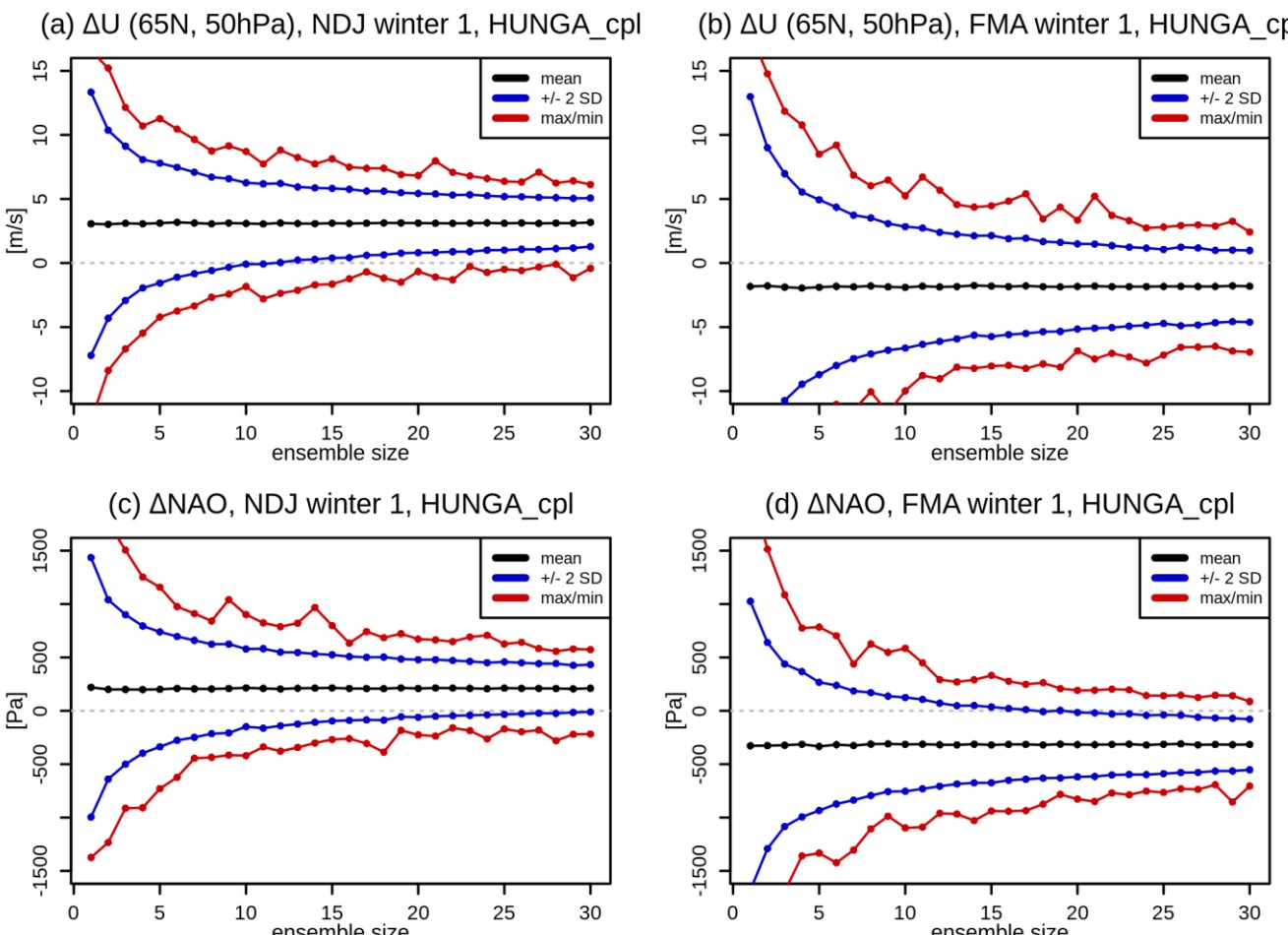

**Figure 9. Detectability of the early winter (a,c) and late winter (b,d) NH stratospheric vortex (a,b) and NAO (c,d) response in the coupled ocean simulations in the first winter following the eruption (2022/2023). Black lines denote the mean response, and blue and red lines indicate the +/- 2 standard deviation and the maximum/minimum ranges, respectively, of the possible responses obtained by randomly subsampling the ensemble with replacement to obtain 2000 artificial ensembles of each different ensemble size. See Fig. S5 for the corresponding changes in the atmosphere-only simulations.**

## 5. Seasonal changes in the SH polar vortex and impacts on the extratropical surface climate

As in the NH, changes in the radiative heating and thus stratospheric temperatures as the result of enhanced aerosol and water vapor following the eruption could drive changes in the Antarctic stratospheric vortex and surface climate. In addition, changes in the SH stratospheric winds are strongly correlated with changes in Antarctic lower stratospheric ozone via ozone-circulation feedbacks, and so any response in ozone can influence the polar vortex evolution (and vice versa). Figure 10 shows the evolution of changes in SH zonal winds at 60°S (a-b) and Antarctic geopotential heights (c-d) for 5 years following the eruption.

No robust change in the Antarctic polar vortex strength is found in the coupled ocean simulations in the first year following the eruption. Note, however, that our diagnostic of polar vortex strength does not capture any changes in jet position, for instance the equatorial shift of winter stratospheric jet discussed in Wang et al. (2023). While the atmosphere-only simulations show some statistically significant strengthening of the stratospheric winds at 60S in austral spring, the response does not propagate down to the troposphere, which shows a negative Southern Annular Mode (SAM, Thompson and

Wallace, 2000) like response in late austral spring and early summer, Fig. 11a, i.e. opposite of what is expected from a strengthened polar vortex. In the second year, however, both simulations suggest a strengthening of the Antarctic vortex in austral spring (i.e. late 2023). In the atmosphere-only simulations the response extends down to the troposphere, and at the surface manifests in sea-level pressure as the positive phase of the SAM - i.e. reduced sea-level pressure over the Antarctic and increased pressure in the SH mid-latitudes (Fig. 11b), with associated cooling over the Antarctic continent and warming

over the southernmost part of South America (Fig. 11e) over late austral spring and early summer (NDJ). Similar surface signatures of a positive SAM like response have also been found in the context of impacts on Antarctic ozone depletion by man-made halogenated substances (e.g. Thomson et al., 2011; Keeble et al., 2014). In agreement, our simulations also show significant SH ozone reductions in the lower stratosphere, primarily in the mid-latitudes (as noted already in e.g. Wang et al., 2023) but also in the springtime Antarctic region (Fig. 12). These negative polar ozone anomalies are driven by anomalous

chemical (chlorine, bromine and nitrogen) processing on aerosol surfaces under conditions of water-induced stratospheric cooling alongside dynamical contributions from altered circulation and ozone transport (Bednarz et al. 2025), and are strongest in year 2 (i.e. 2023) due to the time it takes for Hunga aerosol and water vapour to reach the polar stratosphere (Section 2, Fig. 1). Since ozone absorbs strongly the incoming solar radiation, the reduction in Antarctic ozone from the Hunga eruption cools the polar stratosphere (Fig. S1ab) and further strengthens the polar vortex (Fig. 10a-b) in a two-way

manner, and the stratospheric response then propagates down to the troposphere and affects surface climate.

Notably, in addition to having much stronger stratosphere-troposphere response, the atmosphere-only simulations show substantially larger variability of the Antarctic polar vortex - ozone response (as indicated by the significantly wider spread of potential responses in Fig. 13). This could be because the ENSO changes in the coupled ocean simulations drive SH

stratospheric teleconnection patterns that destructively interfere with the Hunga impacts. In particular, reanalysis and model data show that La Nina-like tropical Pacific anomalies are associated with a weaker SH springtime stratospheric polar vortex and its tropospheric extension (Stone et al. 2022), and this thus agrees with and could explain the much weaker Hunga-induced polar vortex strengthening in the coupled runs. In addition, while the strengthening of the polar vortex associated with the Antarctic ozone reduction in the coupled simulations is also evident in the third winter (i.e. late 2024), the response

is not found in the atmosphere-only simulations (Fig. 10b). This may also be explained by the contribution of ENSO variability in the coupled runs, whereby the now forming El Nino-like response enhances the strengthening of the stratospheric vortex.

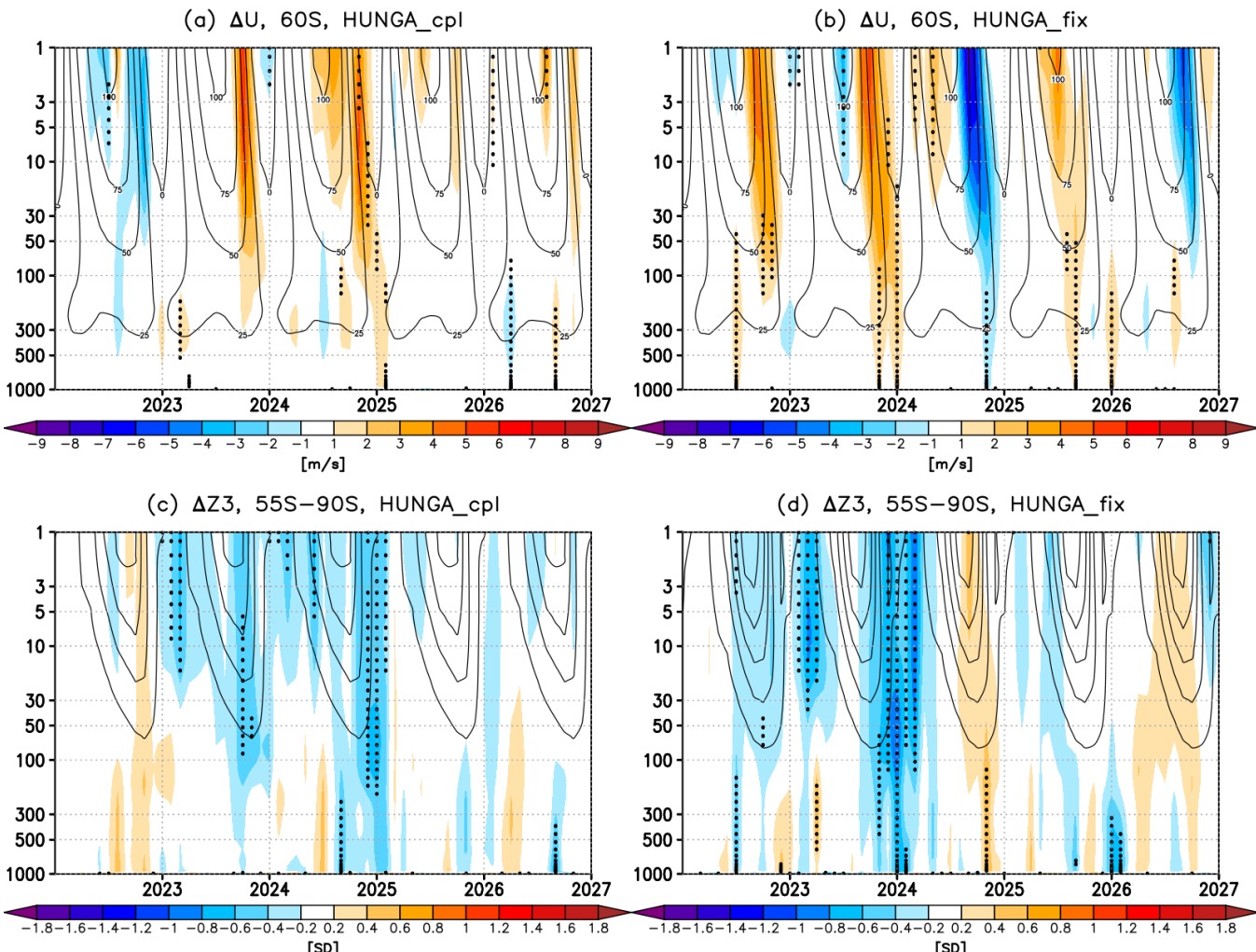

**Figure 10. Evolution of the SH polar vortex. Shading: Timeseries of changes in (a-b) zonal wind at 60°S [in units of m/s] and (c-d) geopotential height [in units of standard deviations] averaged over the Antarctic polar cap (55°S-90°S) between the forced simulation and the control. The left panels are for the coupled ocean simulations and the right panels are for the atmosphere-only simulations. Stippling denotes regions where the response is statistically significant (defined as in Fig. 1).**

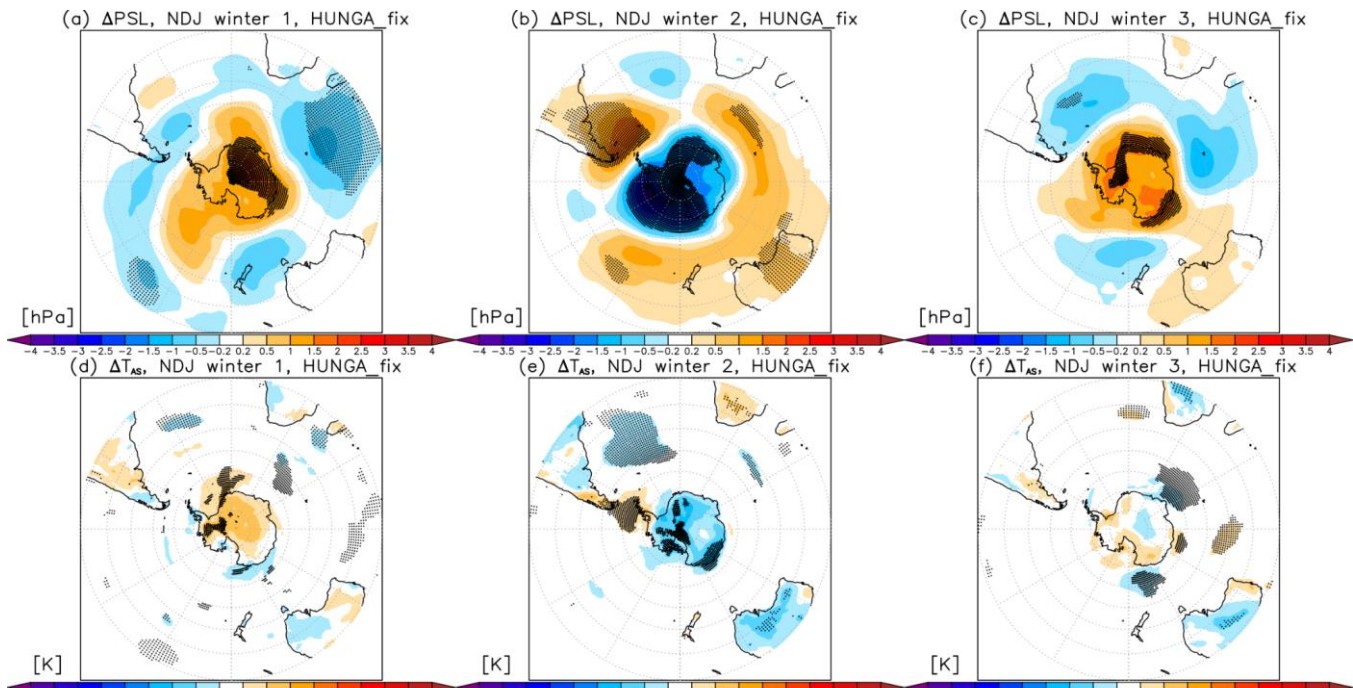

**Figure 11. Changes in late austral spring and early summer (November-December-January) (top) sea-level pressure and (bottom) near-surface air temperature between the forced simulation and the control in the atmosphere-only simulations for the first (i.e. 2022/2023), second (i.e. 2023/2024) and third (2024/2025) spring/summer following the eruption (columns). Stippling denotes statistical significance (defined as in Fig. 1). See Fig. S6 in the Supplement for the corresponding changes in the coupled ocean simulations.**

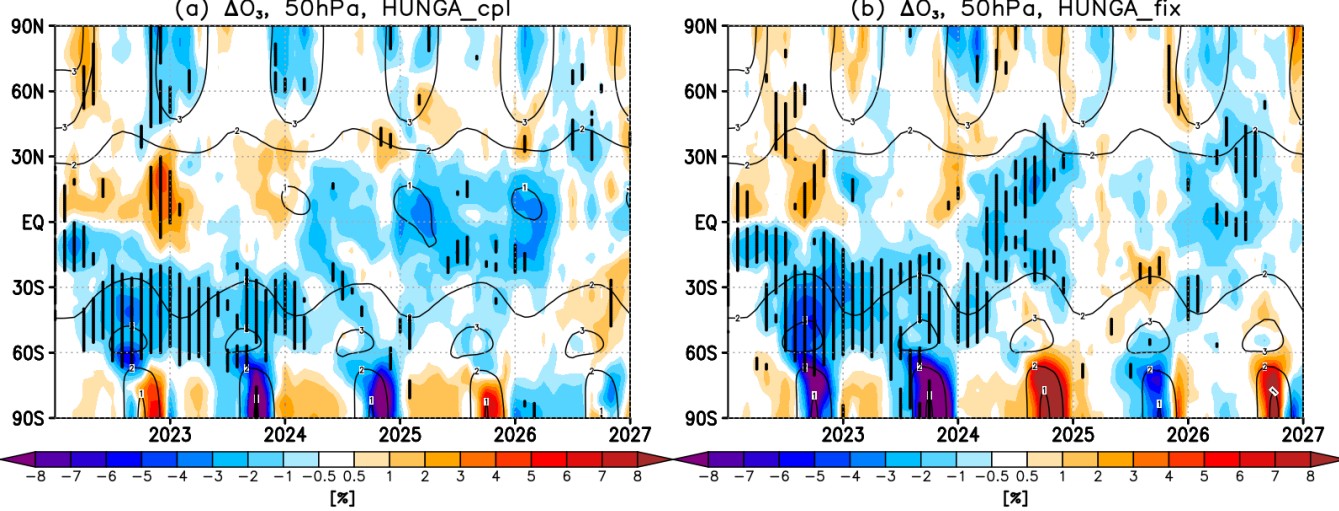

**Figure 12. Timeseries of zonal mean ozone changes at 50 hPa between the forced simulation and the control for the coupled ocean (left) and atmosphere-only (right) simulations. Stippling denotes statistical significance (as in Fig. 1).**

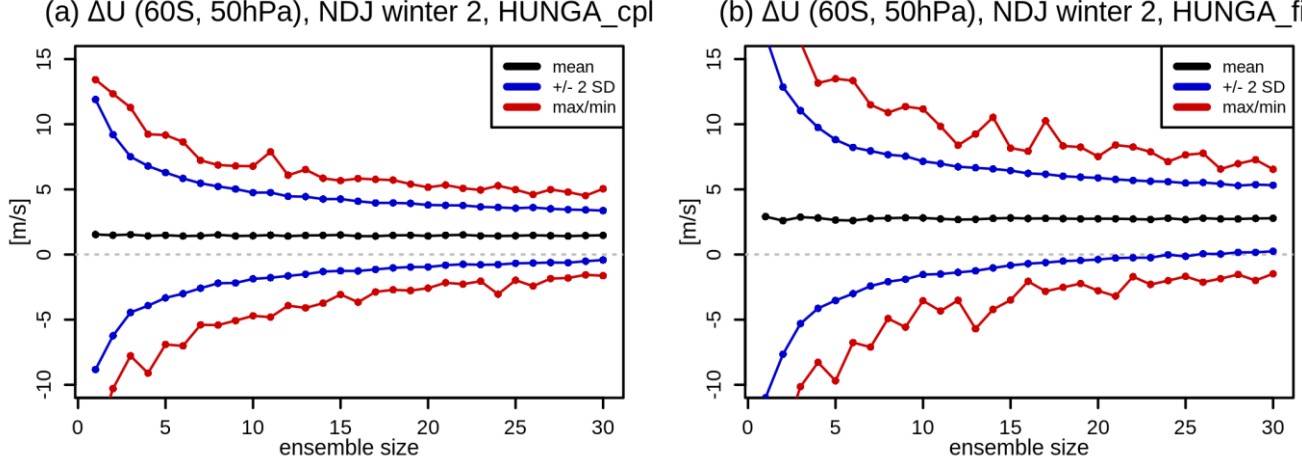

**Figure 13. Detectability of the changes in the 2023/2024 November-December-January SH lower stratospheric vortex in the coupled ocean (a) and atmosphere-only (b) simulations. Different lines as in Fig. 9.**

### 6. Conclusions.

Observations and modelling studies of the Hunga eruption in January 2022 have linked the anomalous stratospheric aerosol and water vapor levels with statistically significant changes in stratospheric and mesospheric temperatures (Stocker, et al., 2024; Wang et al. 2023; Randel et al., 2024; Yu et al., 2023) and ozone (Zhang et al., 2024). In the troposphere, on the other hand, any direct radiatively-driven surface impacts of the eruption are likely to be small (Schoeberl et al., 2024; Quagia, et al. 2025). However, whether the eruption could lead to surface climate changes via other, more indirect pathways remains not well understood. Here we address this using large (30-member each) ensembles of CESM2(WACCM) earth system model simulations with interactive chemistry and aerosols forced both with and without SO2 and H2O injections; the simulations are also carried out using either the atmosphere-only or coupled-ocean configuration to investigate the role of atmosphere-ocean coupling.

We find some statistically significant extratropical circulation and regional climate responses to the Hunga eruption, particularly in the coupled ocean simulations. These arise because of the combination of the ENSO response to the eruption, and its teleconnections, in this model alongside the associated polar vortex changes in both hemispheres. The modulation of ENSO manifests itself in the form of La Nina-like response in years 1-2 (2022-23), followed by an opposite, El Nino-like response in year 4 (2025). The La Nina-like response following the eruption is consistent with that previously inferred for the past SH volcanic eruptions (Pausata et al. 2020, 2023; Ward et al. 2021) and SH wildfire aerosols (Fasullo et al. 2023), suggesting changes in the ITCZ following a hemispherically unsymmetrical forcing could be the main driver contributing to

the ENSO response here. Our results also suggest that the eruption could have contributed to the anomalous persistence of the La Nina-like conditions observed between 2021-2023 (e.g. Iwakiri et. al., 2023). The modulation of the ENSO variability

following the eruption in the model also gives rise to a small global mean cooling around the same period, as well as a number of regional surface temperature responses in the Pacific region that are generally consistent with those associated with anomalous Pacific SSTs and their teleconnections more generally (e.g. Domeisen et al. 2019). In contrast to earlier results of Jucker et al. (2024), we do not find any prolonged robust regional surface climate responses in the atmosphere-only simulations, especially in the later parts of the 10-year-long simulations when most of the anomalous aerosol and water

vapor is removed from the stratosphere.

The simulations also show some statistically significant stratospheric polar vortex responses in both hemispheres. These changes propagate down to the troposphere and affect surface climate. In the NH, these surface climate responses manifest themselves as modulations of the North Atlantic Oscillation. The NH extratropical responses change sign between early and

late winter, with generally stronger polar vortex and positive NAO-like response at the surface in early winter, and vice versa for late winter, and are simulated during three boreal winters following the eruption in the coupled run. The modulation of the polar vortex under the Hunga eruption could be driven by the changes in radiative heating and cooling under enhancement of stratospheric aerosols and water vapour. In such case,  increased stratospheric aerosols  act to increase tropical lower stratospheric temperatures and meridional temperature gradients and hence  strengthen lower stratospheric

polar vortex, in a manner suggested for past sulfur-rich eruptions (e.g. Polvani et al., 2019). In contrast,  increased $H2O$ cools mid-to-upper stratosphere and modulates meridional temperature gradients at these altitudes, thereby potentially weakening polar vortex higher up; it was suggested that the initial upper stratospheric vortex anomaly could propagate down to the lower stratosphere and affect the troposphere in spring (Kuchar et al. 2025). In addition, the simulated changes in the polar vortex, especially in the NH, are likely to arise not only because of radiative/temperature changes in the stratosphere

due to the aerosols and water vapor, but also at least partly due to the modulation of ENSO and the associated changes in tropospheric wave flux to the stratosphere. In accord, the atmosphere-only simulations show NH polar vortex changes that are significantly weaker and have different seasonality than those in the coupled ocean runs, and largely do not propagate down to the troposphere and affect surface climate. This suggests that the modulation of ENSO and its teleconnections plays an important contribution to the polar vortex changes simulated after the eruption.


For the SH response, the simulations show strengthening of the SH polar vortex in austral spring of 2023 that is likely at least partly related to the associated reduction in Antarctic ozone as increased stratospheric aerosols and water vapour reach the polar vortex. In the atmosphere-only case, the stratospheric response propagates down to the surface, leading to the pattern of changes in sea-level pressure projecting on the positive phase of the Southern Annular Mode, and some

statistically significant near-surface temperature changes, including a small cooling over Antarctica. However, here the influence of the ENSO modulation on the vortex in the coupled ocean runs likely acts deconstructively with the impacts

driven directly in the stratosphere, and so the coupled ocean simulations show reduced variability of the polar vortex and its weaker response to the eruption than that found in the atmosphere-only simulations. A schematic representation of the potential indirect pathways of Hunga impacts on regional climate is included in Fig. 14.


Finally, we examine the role of interannual variability and ensemble size for the detectability of the climate signals from the eruption. By randomly sub-sampling the ensemble we show the range of apparent surface climate responses that can be inferred if smaller ensemble sizes are used. We demonstrate that interannual variability is likely to have a first order influence on the response inferred from an ensemble of smaller (e.g. 10 members) size, but that it can still have a non-

negligible contribution even if as much as 30 members are used. The results thus highlight the need for caution when interpreting the surface climate impacts of the eruption inferred from insufficient ensemble size.

All in all, our study suggests that the Hunga eruption could have a non-negligible influence on regional surface climate, and discusses the mechanism via which such an influence could occur. However, the results also highlight that this forcing is

relatively weak compared to interannual variability, is subject to model uncertainties in the representation of key processes (e.g. it is unclear whether the ENSO response is particular to the model sensitivities) as well as in some parts requires a large number of ensemble members to confidently detect. This calls into question whether such an influence could be robustly detected in the single 'realization' of the real world. Still, studies show that current climate models tend to underestimate the signal-to-noise ratio and the predictable component of the forced response regionally (e.g. Scaife and Smith, 2018; Gillet et

al., 2003; Blackport and Fyfe, 2022; Williams et al.2023; Smith et al., 2025), suggesting that the indirect climate response to the eruption seen in this study may not adequately represent the real world response. More research is thus needed before definitive statements on the role of the eruption in contributing to the surface climate and weather events in the following years are made.

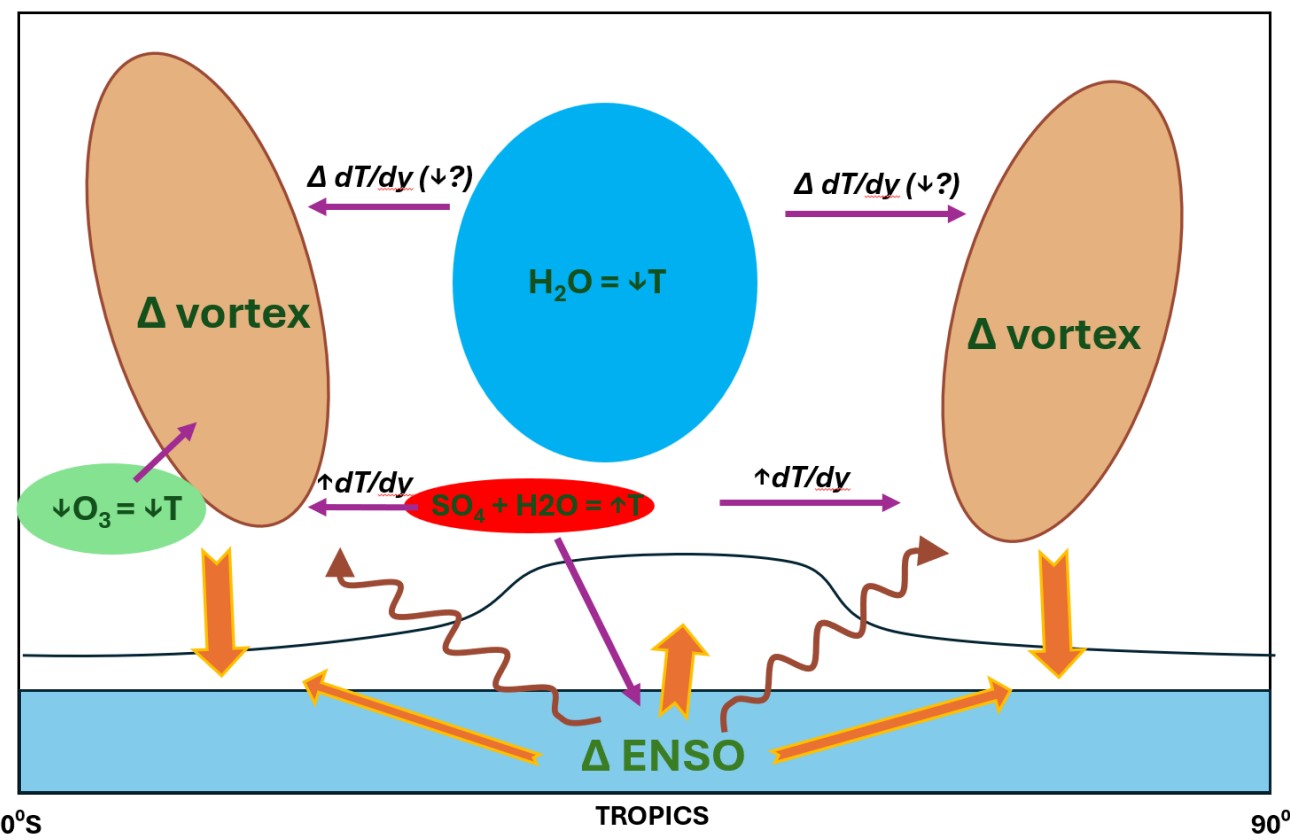

**Figure 14. Schematic representation of potential indirect pathways of Hunga influences on regional climate. Asymmetric volcanic forcing can influence ENSO variability, which modulates tropospheric climate in both the tropics and at higher latitudes via teleconnections, as well as influences planetary wave activity which impacts stratospheric winds. Modulations of the polar vortices are also possible via changes in stratospheric meridional temperature gradients (which can vary in sign with altitude due to differences in temperature perturbations from aerosols and water vapour) and, in the southern hemisphere, Hunga impacts on the Antarctic ozone. These stratospheric polar vortex changes can propagate down to the troposphere and affect surface climate via modulations of extratropical modes of variability (NAO in the NH, SAM in the SH).**

**Acknowledgements**

We would like to acknowledge high-performance computing support from Derecho and Casper, provided by the National Center for Atmospheric Research (NCAR) Computational and Information Systems Laboratory, sponsored by the National Science Foundation.

Support has been provided by the National Oceanic and Atmospheric Administration (NOAA) cooperative agreement NA22OAR4320151, NOAA Earth Radiative Budget (ERB) program and the Reflective followship program. WY's work is prepared by LLNL under Contract   DE-AC52-07NA27344.

We thank Simone Tilmes for help setting up the simulations, and Mark Schoeberl for helpful comments on the study.

**Authors Contributions**

EMB conceptualized the study, analysed the data and wrote the manuscript, with the support from AHB. XW, EMB, ZZ and WY performed the simulations. XW and YZ designed the simulations. All authors contributed to discussion of the results and writing of the manuscript.

**Conflict of Interest**

EMB and MT are members of the editorial board of ACP.

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
