# Peer review of "Indirect climate impacts of the Hunga eruption"

_EGUsphere, 2025_

## Community Comment (CC1)

Dear authors,

we appreciate the manuscript by Ewa Bednarz and colleagues and we are convinced it will make an interesting contribution to the Hunga-Tonga literature. After having gone through the manuscript in detail, we wish to encourage some additional analysis with respect to Kuchar et al (2025). We would encourage the authors to examine in more detail the atmospheric-only simulations (HUNGA_fix) even though the climatological SSTs may dampen the surface response.

It appears that HUNGA_fix has the potential to represent the top-down mechanism suggested in Kuchar et al (2025), i.e. the influence of lower-mesosphere cooling on the stratospheric polar vortex via decreased temperature gradient. Because of HTHH-MOC (Zhu et al., 2024), there is access to the presented simulations which allowed us to reproduce Fig. 2 from Kuchar et al (2025), see Fig. C1 below. As you can see most of the features such as the mesospheric cooling at lower latitudes (Fig. C1a), which has the potential via the thermal wind relation to weaken the polar-night jet, are represented by the HUNGA_fix simulation. Consequently, we observed an increase in polar temperature progressing downward and the associated cooling aloft (Fig. C1b). However, the statistical significance at higher latitudes is rather limited possibly due to the use of monthly means. Note that Fig. C1 shows the evolution of temperature for the winter 2023/2024 in contrast to 2022/2023 used in SOCOLv4 in Kuchar et al (2025). As discussed therein, the mechanism suggested and simulated by SOCOLv4 above should be valid for the winter of 2023/2024 and the following winters if the lower-mesospheric cooling is persistent and strong enough due to the excess WV. This difference has been attributed to the overly fast transport of water vapour in SOCOLv4 and differences in the experiment protocol.

We also reproduced Fig. 9 to assess the detectability of the NH stratospheric vortex in the early winter (Fig. C2a), late winter (Fig. C2b) and March only (Fig. C2c) in SOCOLv4. Similarly, we cannot detect any statistically significant signal beyond the envelope of potential responses in the early and late winter based on our sample size. However, when we assess the responses only in March 2023 we see an emerging negative response in zonal-mean zonal wind at 65°N and 50 hPa. This suggests that emerging signals on a monthly basis may be detected even with a limited sample size. Accordingly, we would like to suggest further exploring the detectability of signals for the winter 2023/2024 as done in Figs. 9 and S5.

Best regards

Ales Kuchar, Timofei Sukhodolov, Eugene Rozanov, Gabriel Chiodo, Harald Rieder

[Figure]

**Figure C1** *Weighted zonally averaged temperature averaged over 0 -20° N (a) and 60 -90° N (b) monthly anomalies for 202312-202405 in WACCM (HUNGA_fix). The anomalies are expressed as the differences between the simulations with and without HT forcing. The 2σ statistical significance from a t test is indicated by the dots.*

[Figure]

**Figure C2** *Detectability of the NH stratospheric vortex in the early winter (A), late winter (B) and March only (C) in SOCOLv4 based on daily means. Error bars represent +/- 2 standard deviation of the possible responses obtained by randomly subsampling the ensemble with replacement to obtain 2000 artificial samples of each different ensemble size.*

**References**

Kuchar, A., Sukhodolov, T., Chiodo, G., Jörimann, A., Kult-Herdin, J., Rozanov, E., and Rieder, H. H.: Modulation of the northern polar vortex by the Hunga Tonga–Hunga Ha'apai eruption and the associated surface response, Atmos. Chem. Phys., 25, 3623–3634, https://doi.org/10.5194/acp-25-3623-2025, 2025.

Zhu, Y., Akiyoshi, H., Aquila, V., Asher, E., Bednarz, E. M., Bekki, S., Brühl, C., Butler, A. H., Case, P., Chabrillat, S., Chiodo, G., Clyne, M., Falletti, L., Colarco, P. R., Fleming, E., Jörimann, A., Kovilakam, M., Koren, G., Kuchar, A., Lebas, N., Liang, Q., Liu, C.-C., Mann, G., Manyin, M., Marchand, M., Morgenstern, O., Newman, P., Oman, L. D., Østerstrøm, F. F., Peng, Y., Plummer, D., Quaglia, I., Randel, W., Rémy, S., Sekiya, T., Steenrod, S., Sukhodolov, T., Tilmes, S., Tsigaridis, K., Ueyama, R., Visioni, D., Wang, X., Watanabe, S., Yamashita, Y., Yu, P., Yu, W., Zhang, J., and Zhuo, Z.: Hunga Tonga-Hunga Ha'apai Volcano Impact Model Observation Comparison (HTHH-MOC) Project: Experiment Protocol and Model Descriptions, EGUsphere [preprint], https://doi.org/10.5194/egusphere-2024-3412, 2024.

---

## Author Comment (AC1)

**Authors Response to Reviewers**

Reviewer comments – black

Authors response – blue

**Reviewer 1**

This work by Bednarz et al. studies the climate impacts of the dramatic and strong Hunga eruption, where large quantities (150Tg) of H2O was injected into the stratosphere using CESM2 (WACCM6) ensemble (30 members). Much less SO2 was in the eruptive material, 0,5-1,0Tg, so this case study is relevant when it comes to understand how the optical properties of water can affect various atmospheric thermal gradients and thus climate, compared to SO2 that is usually in the focus. Their findings include a sigificant La Nina response in the first two years following the eruption that in turn trigger changes in the NAO via changes in tropospheric wave flux up into the stratosphere. SH was also explored where changes in the SAM was also identified. In general the results are quite convincing and where the possible mechanistic pathways are detailed. This is a well written manuscript and I do recommend it's publication in ACP after addressing a few comments that I have.

We thank the reviewer for the positive review and helpful comments which we address below.

First, I wonder if adding a figure showing the temperature response with time from the surface and up (like Figure 5/Hovmoller plot), for say 0-20°and 65-90°. That could show more clearly how strongly the eruption impacts the stratosphere and the associated climate impacts.

We agree that having plots of the associated temperature changes is important, but note we already show the latitude vs time evolution of temperature changes in the lower (50 hPa), middle (25 hPa) and upper stratosphere (1 hPa) in the supplement (Fig. S1). To make this more clear, we have now added the reference to this plot already in the section discussing the simulated AOD and H2O changes. Regarding the polar temperature changes, our Figs. 5 and 10 include changes in geopotential heights over the polar regions, which are essentially proxy for the associated polar temperature changes.

L40-41: Here more explanation would be good, since the high AOD is basically due to the large stratospheric water content that enhances the reaction of SO2 into sulfate aerosols in addition to supporting the growth of the sulfate particles themselfs.

We agree and have added the following explanation: "partly due to the presence of anomalous water vapor enhancing aerosol growth in the initial months following the eruption (Zhu et al., 2022; Quaglia et al., 2025)."

L56: It would be good to mention the water vapor lifetime in the models vs observations/measurements, how realistic is it to expect such a delayed response due to water vapor only? Not in the study of Jucker et al but more in general.

We agree and have added the following sentence to that paragraph: "With the Hunga water stratospheric burden e-folding time of ~31-43 months estimated from the current chemistry-climate models (Zhuo et al. 2025), the reasons behind such a significant delay in the emergence of the response are also not well understood."

L121: „Some significant, largely negative…" sounds confusing, so these significant anomalies are they largely negative?

That's correct; we have now rephrased this to 'some significant and mostly negative'

L124: this is section 4, right?

Apologies for the typo; however, we have now changed the section numbering (incorporating the former Section 3 into Section 2.3), and so Section 4 here is now correct. We apologize for the confusion.

L126: Section 5 perhaps?

As above, we apologize for the typo and confusion. Section 4 as it states is now correct.

Figure 1: So even where the anomalies are zero (white), the response is significant? Please explain.

White shading in our plot does not indicate the anomalies are 0, but between -1e-3 to 1e-3 for AOD and between -0.2 and 0.2 ppm for H2O. While these values are much smaller than the peak anomalies in the plot, they are still (statistically) significantly higher than the values in the control experiment.

L210-211: I think it needs mentioning that such a mechanism has been proposed with respect to sulfate-rich stratospheric eruptions – but here the focus is mainly on a water vapor-rich eruption. I think that the small amount of SO2 causes a negligible impact on the PV here, despite the amplification/growth of SO2 due to water vapor.

We agree and have modified the sentence to read "… for previous **sulfur-rich** volcanic eruptions (Polvani et al., 2019; Paik et al., 2023), although some studies point that this polar vortex signal only emerges for eruptions with large aerosol loading (Azoulay et al. 2021; DallaSanta and Polvani, 2022**), much larger than the 0.5 Tg SO2 injected in these simulations.**". The fact we focus here on a water-rich eruption comes in the next sentence: "However, a unique aspect of the Hunga eruption was the exceptional water vapor injection."

The authors use single model large ensemble simulations to assess the potential impacts of the 2022 Hunga Tonga volcanic eruption. The manuscript is generally well written and presents interesting results well worthy of publication.

We thank the reviewer for the positive review and helpful comments which we address below.

My main comment is that there is a certain lack of acknowledgment that many of the described results have been reported earlier. There is ample referencing in the introduction, but the results are mostly presented as new findings, while to me they are mostly confirmations of previous findings, or more detailed but similar results. In particular, the cited Jucker et al (2024) study is very similar in several points: It uses WACCM, has 30 ensemble members , discusses longer-term surface impacts, shows a figure very similar to Fig 1c,d, and discusses a similar global wave train and a potential role of changes in ENSO.

Of course, that study only adds water vapor and does not start from the observed state of the atmosphere. So the current simulations are certainly different, but not to a point to be entirely new.

We agree with the reviewer that the manuscript would benefit from more direct comparisons with previous results, in particular the Jucker et al. (2024) and the Kuchar et al (2025) studies, and we have now added more of such direct comparisons.

However, we note that our results, while also using the WACCM model, differ significantly from that of Jucker et al., not only by imposing both aerosol and water perturbations in conjunction as well as initializing from observed conditions, but even more importantly by using two WACCM configurations, one with fixed SSTs and one with coupled ocean, while Jucker et al used only fixed SSTs WACCM runs but complemented with additional idealized MiMA simulations with interactive mixed layer ocean. In addition, we analyzed extensively changes in stratospheric circulation and stratosphere-coupling induced by the eruption, as well as detectability of some of the indirect climate responses, which are aspects not covered in the Jucker et al. study.

In addition, this manuscript seems to be a spin-off from the APARC Hunga Tonga impact comparison effort, from which there will probably be more publications forthcoming. How is this study then different to others? For instance, I noticed another submitted manuscript, the cited Zhuo et al (2025) has some common authors and shows an almost identical Figure 1. So similar to my comment above, I believe this should be acknowledged more openly in the manuscript, and more context given.

Zhuo et al. (2025) analyzed changes in AOD as well as global-mean evolution of stratospheric $H_2O$, T and $O_3$ in the HTHH-MOC models and compared them with observations. Our study focuses on the Hunga impacts on tropospheric temperatures, ENSO teleconnection, stratospheric circulation and stratosphere-troposphere coupling (using a single model from the HTHH-MOC, albeit one run in two different configurations and consisting of significantly more ensemble members than the other models). While our Fig.1,

showing simulated evolution of AOD and $H_2O$, is similar to the results in Zhuo et al. (2025), we show it here to illustrate the forcing used in these simulations and so to give a better context for the main results presented in later parts of the manuscript (rather than to show 'new' results). In addition, we clearly acknowledge the Zhuo et al. 2025 results in that section: "The evolution of aerosols and water vapor simulated in CESM2(WACCM) is thus similar to that simulated by other models participating in the HTHH-MOC and inferred from available satellite data (Zhuo et al., 2025)". Nonetheless, to make it clear the results in Section 3, Fig. 1., are used mostly for context and to illustrate the drivers of the climate responses studied later, we have now moved Section 3 into the Methods section (now Section 2.3).

In terms of how the paper fits into wider analysis of the HTHH-MOC, we already mention the introductory paper by Zhu et al. (2025) discussing the protocols and models, make references to the above discussed Zhuo et al. (2025) paper that analyses the evolution of AOD, global mean stratospheric H2O, temperatures and ozone as well as to the recent Quaglia et al. (2025) paper that discussed the resulting Hunga impacts on the atmospheric radiative budget and surface forcing. We're unaware of any other published papers utilizing the results from the HTHH-MOC.

To be clear, I wouldn't want this manuscript to become a constant comparison to other papers. Just giving more of an idea of where these results lie with respect to published literature would be good.

We agree and have added references and comparisons with the previous works when relevant.

**More specific comments:**

L 28: "demonstrates": I am not sure one should be that definite. There are still open questions and as noted above, others have reported similar results.

We agree and have changed this to "suggests".

L34: "eruption erupted" - not wrong, but maybe one can find a more elegant expression?

Thank you for spotting this – we have changes this to "volcano erupted"

L81: "Meinshausen", not "Meineshausen"

Corrected.

L82-83: I think the way the ocean is initialized is rather important. So even if it is described in Richter et al (2022), I think it would be good to describe this in more detail. In particular, how do the authors deal with model drift which is often the problem in coupled simulations initialized close to observation?

We acknowledge that initializing the model from observed values leads to some model drift in the first few years as the model moves away from the imposed observed state towards its own quasi-equilibrium state. However, since exactly the same initial conditions are used both in the control and perturbed simulations, such a drift is present in both runs. We always analyze the Hunga responses by taking the difference between perturbed and control (rather

than looking at absolute values), thereby to the first order removing any impacts such drift could have on the results. We now state this in the methods sections 2.2:

"We note that initialization of the model with the observed conditions leads to some model drift in the first few years as the model moves away from the imposed observed conditions towards its own quasi-equilibrium state; this is particularly the case for the ocean component of the coupled runs (now shown). However, since exactly the same initial conditions are used both in the control and perturbed experiment, and the Hunga response is always taken as the difference between perturbed and control (rather than looking at absolute values), this to a first order removes any impacts such drift may have on the inferred results. "

L83-84: "first 1-2 months": I understand this depends on ensemble member as the nudging is how the ensemble is created. But this is not clear at this point at all. Consider adding just a little but more information, as my immediate reaction was "so what is it, 1 or 2 months?"

Thank you for highlighting this – we have now clarified this to "the first 1-2 months, depending on the ensemble member (see below), .."

L116, Figures 2,3: Why did the authors decide to focus on annual means versus seasonal impacts? Warm versus cold season impacts could be rather different.

This is partly because we discuss the seasonal impacts (especially in years 1-3) in sections 4-5, and we wanted to avoid repetition. In addition, while we agree that annual means vs seasonal impacts could be different, we have previously plotted figures similar to Fig. 2-3 but for DJF and JJA only (see Fig. R1-R4 below), and the main conclusions were not substantially different to those reached with annual means – i.e. that no robust longer-term changes can be found in the atmosphere-only simulations, while the changes in the coupled ocean runs are largely indicative of any associated changes in ENSO teleconnections, polar vortex and stratosphere-troposphere coupling. We now add a sentence about this to Section 3: "Broadly similar conclusions are reached if seasonal mean responses are used instead of annual means (not shown)."

[Figure]

*Figure R1. December-January-February mean changes in near-surface air temperature between the forced simulation and the control in the coupled ocean simulations in each of the 9 years following the eruption (with DJF-1 indicating December 2022 to February 2023 response, DJF-2 indicating December 2023 to February 2024 response, etc) Stippling indicates statistical significance (defined as in Fig. 1).*

[Figure]

*Figure R2. As in Fig. R1 but for the responses in the atmosphere-only simulations.*

[Figure]

Figure R3. June-July-August mean changes in near-surface air temperature between the forced simulation and the control in the coupled ocean simulations in each of the 9 years following the eruption (with JJA-1 indicating JJA 2022, JJA-2 indicating JJA 2023 response, etc) Stippling indicates statistical significance (defined as in Fig. 1).

[Figure]

*Figure R4. As in Fig. R3 but for the responses in the atmosphere-only simulations.*

L127: This already is section 4. Maybe this should be "Section 5"?

Apologies for the typo; however, we have now changed the section numbering (incorporating the former Section 3 into Section 2.3), and so Section 4 here is now correct. We apologize for the confusion.

L139-146: Why did the authors decide to focus on the early La Nina response and largely discard the later El Nino response? Could it be that La Nina is forced by aerosols and El Nino by water vapor in the long term given its longer residence time? Or could it be that the coupled model shifts into an El Nino as a response to the La Nina, regardless of longer-term forcing? Related to this, why did the authors decide to stop the analysis after 5 years even though their simulations were run for 10 years and the signals are still pretty strong in year 5?

We note we didn't stop the analysis entirely after 5 years, and in fact we do have similar plots to Fig. 2-3 bit for the annual mean near-surface temperature changes in years 6-10 in the supplement (Figs. S2-S3); we now refer to them more clearly in the manuscript and discuss them too.

We think that the fact the model shifts from the La Nina like response in Y1-2 to El Nino like response in Y4 (and also, as now pointed out in the manuscript, back to La Nina like response in Y9) is not necessarily indicative of long-term changes in the Hunga forcing itself but rather the long-term result of the initial perturbation and how it influence the oscillatory nature of ENSO in the model in the long-term. Given that there are many uncertainties in the details of such long-term perturbation, including the exact timing of thereof, we chose not to put too much stress on the ENSO changes in later parts of the simulations. But we now acknowledge all these points more clearly in the manuscript:

"The role of potential Hunga modulation of the ENSO variability and its teleconnections in contributing to some of the surface climate responses, as found here for the coupled ocean simulations (Fig. 2 and S2), has also been suggested by Jucker et al. (2024) using a second set of experiments with a medium complexity general circulation model MiMa. In their case, however, the response resembled a positive ENSO – or El Nino-like – response in years 4-9, unlike the La Nina-like response in year 1-2 followed by El-Nino-like response in year 4 here (as well as a La-Nina-like response again in year 9; see Fig. S2). The simulated ENSO changes in later parts of our simulations are likely not indicative of long-term changes in the Hunga forcing itself (which weakens over time) but rather are the long-term result of the initial perturbation and how it influences the oscillatory nature of ocean-atmosphere ENSO feedback in the long-term. Given that ENSO is not perfectly periodic and there are many uncertainties in the details of such long-term modulation due to stochastic noise influencing these different feedbacks at different times (e.g. Wang, 2001), there are many uncertainties in the exact timing of the different responses in the model, and so we focus here mostly on the results in the initial few years following the eruption."

L160-161: I agree, but the fixed SST simulation still has wave-like signals in the extratropics. Where do those come from?

Atmospheric internal variability can also manifest itself in the form of wave-like structures, including, e.g. stratosphere-troposphere coupling, even when ocean boundary conditions are fixed.

L166: this should probably be "(section 5)" and "(section 6)"

As above, we apologize for the typo; however, we have now changed the section numbering (incorporating the former Section 3 into Section 2.3), and so Section 4 and 5 here is now correct. We apologize for the confusion.

L233-234: I agree, but one could also invert the argument that if there are surface signatures found even in the fixed SST simulations, it is likely that the real impact (where the ocean is coupled) should be expected to be larger.

If ocean feedbacks act to merely amplify the initial atmosphere perturbation then yes, the real impact would likely be larger. But if ocean feedbacks act to oppose the atmosphere perturbation (for instance because of the concurrent impacts on ENSO and the associated teleconnections dominating over the local top-down response from the atmosphere) then the real perturbation would not necessarily resemble that in the fixed SSTs runs. At any rate,

we do not claim that the atmosphere-only runs cannot provide any helpful understanding of the responses in the real world, but simply that the results need to be interpreted carefully.

We have rephrased: "... highlighting the need for caution when interpreting the inferred signatures, or their absence, of the Hunga eruption on climate." to read "highlighting that interpreting the inferred signatures, or their absence, of the Hunga eruption on climate using atmosphere-only model simulations needs to be done carefully." to make this clearer.

Figure 5: Again, there are still significant signals at the end of 2026, so why stop here? How does this look for 2027 and later?

We have redone Fig. 5 but for the period 2027-2031 (Fig. R5, below). There are no clear robust anomalies, other than some modulations of the polar vortex in the coupled run associated with the concurrent modulation of ENSO; as discussed above, given the uncertainties in the model long-term ENSO response, we prefer to focus the analysis on the period 2022-2026.

[Figure]

**Figure R5.** *As in Fig. 5 of the main manuscript bot for the responses in years 6-10 (2027-2031).*

L381: Domeisen, not Domaisen

Corrected.

L389-391: This is a very long parenthesis. If there's so much to say, it's probably better to say it in a separate sentence.

We agree and have split this into separate sentences.

L416: Again, I am not sure this study "demonstrates".

We have changed this to "suggests".

This manuscript examines the indirect surface climate effects of the January 2022 Hunga volcanic eruption using a large-ensemble simulation approach. The authors employ the state-of-the-art CESM2(WACCM6) Earth system model with interactive chemistry and aerosols in both atmosphere-only and coupled ocean–atmosphere configurations. The 30 ensemble members are used for each configuration, a large sample that improves eruption signal detection relative to previous studies (e.g., 10-member ensembles in the HTHH-MOC project). The study shows that, despite the eruption's weak direct radiative forcing, indirect mechanisms produce statistically significant regional surface climate effects. Model results indicate a La Nina-like cooling of the equatorial Pacific during the first 1–2 years post-eruption, followed by an El Nino-like state around year 4. These ENSO phase shifts subsequently influence extratropical circulation: the North Atlantic Oscillation is altered during boreal winters, and the Southern Annular Mode is modified in austral spring, through a combination of ENSO teleconnections and stratospheric polar-vortex disturbances.

Overall, the manuscript is well-written, and the analysis is comprehensive. The authors effectively use the two model configurations to distinguish the role of ocean coupling, showing that the coupled runs exhibit more pronounced surface impacts (due to ENSO responses) than the atmosphere-only runs. The inclusion of interactive chemistry/aerosols and the consideration of both Northern and Southern Hemisphere effects (including stratospheric ozone changes) are strengths of the study. The manuscript also acknowledges the modest magnitude of the signals relative to internal variability, and it employs statistical significance tests and ensemble subsampling to assess result robustness.

I recommend publication after the following major points and minor fixes are addressed.

We thank the reviewer for the positive review and helpful comments which we address below.

Major comments:

1. The study relies on a single climate model (CESM2(WACCM6)). While this model is sophisticated and the large ensemble lends confidence to internal consistency, it is important to situate the findings in the context of other models and studies. The authors are encouraged to expand the discussion of how their results compare to previous Hunga-Tonga modeling efforts. For example, Jucker et al. (2024) used chemistry–climate model simulations with water-vapor-only forcing and also found multi-year surface responses; the authors might comment on similarities or differences in timing (Jucker et al. saw responses 3–8 years post-eruption) or in mechanisms. A brief comparison would also acknowledge the multi-model context and underscore which findings are novel versus confirmations of prior work.

We agree with the reviewer that our manuscript would benefit for more direct comparisons, and we now include more of such comparisons - especially with the Jucker et al. 2024 and Kuchar et al. (2025) studies, which to our knowledge are the two main published studies to date exploring issues relating to indirect climate impacts of Hunga – into the manuscript, both in the results sections and in the conclusions.

2. The coupled-ocean simulations show significant cooling in Northern Hemisphere extratropical surface temperatures during the first 1–3 years, which the authors attribute to circulation changes rather than direct radiative forcing (since most aerosols remained in the Southern Hemisphere). To strengthen this attribution, it'd be very helpful to analyze and/or clarify the TOA radiative flux changes due to the eruption. For example, the authors could include a decomposition of the TOA radiative perturbation by component (shortwave vs longwave, or aerosol vs water vapor vs ozone contributions) to confirm that direct radiative forcing in the NH is minimal, and thus the NH cooling must stem from indirect effects. It would strengthen the case that circulation, not local direct radiative forcing, drives the NH response.

We agree with the reviewer that this is important. The analysis of the corresponding radiative flux changes (both TOA and surface), including decomposition by component, from these simulations as well as from other models constituting part of the HTHH-MOC is the focus of the paper by Quaglia et al. (2025), which is now in review for ACP and available as a pre-print (https://doi.org/10.5194/egusphere-2025-3769). We now include references to these results when relevant in our manuscript, including the section discussing the significant extra-tropical NH surface cooling in years 1-3.

3. Given the central role of the ENSO response in this study, the manuscript should more directly compare the simulated ENSO behavior to observations. The model predicts a La Nina–like state in 2022–2023 followed by an El Nino–like phase by 2025. In reality, a protracted La Nina persisted from 2021 through late 2022, and an El Nino event emerged by mid-2023 (much of the world experienced a transition to El Nino conditions by late 2023). The manuscript already notes that the eruption may have reinforced the prolonged La Niña episode, but the model–observation comparison could be improved. Explicitly point out that the simulated La Nina-like cooling in 2022–2023 coincides with the observed La Nina, whereas the model's El Nino-like warming appears in year 4, about a year later than the real-world phase shift in 2023–2024. Discuss whether this lag is simply internal variability or whether it indicates that the volcanic perturbation prolonged La Nina conditions.

We note we do state in the Conclusions that "Our results also suggest that the eruption could have contributed to the anomalous persistence of the La Nina-like conditions observed between 2021-2023 (e.g. Iwakiri et. al., 2023)." But we agree with the reviewer that more discussion of this would be useful in the results section, and so we also include such discussion in Section 3.

4. The simulation setup injects 0.5 Tg $SO_2$ and 150 Tg $H_2O$ between 25–30 km altitude on 15 January 2022. The authors should justify the assumed injection altitude profile, and discuss how it might affect the results. For instance, was 25–30 km chosen based on the bulk of the eruption mass being in the lower stratosphere, or to align with the HTHH-MOC protocol? Showing a modelled time-height cross-section for Figure 1 would clarify how the stratospheric aerosol/water vapor spreads vertically

in your simulations. Ideally, a sensitivity test could show that slightly different injection height assumptions would not qualitatively change the outcomes.

First, we've discovered a mistake in the manuscript – the injection actually occurred at 20-28 km for SO2 (with 71% at 20-22 km and 29% at 22-28 km) and at 25-35 km for H2O (with 69% at 25-27 km, 28% at 27-30 km and 5% at 30-35 km). We have now corrected this in the text. Second, our assumed injection profile has been taken from the study of Zhu et a. (2022) where it was chosen based on the evaluation of model results with the observations. We now include the reference to this study in our paper.

The revised text now reads: "For each experiment, there are two pairs of simulations: a 'forced' simulation with simultaneous injection of 0.5 Tg SO2 at 20-28 km (with 71% of injection at 20-22 km and 29% of injection at 22-28 km) and 150 Tg H2O at 25-35 km (with 69% at 25-27 km, 28% at 27-30 km and 5% of 30-35 km) at $22^0$S-$14^0$S and $182^0$E-$186^0$E on 15 January, and a 'control' simulation without the Hunga injection. Such altitude profile of the injections has been chosen based on the results of Zhu et al. (2022), where it was found to produce a relatively good agreement with observed distribution of Hunga water and aerosol, albeit with the SO2 injection scaled to give the total of 0.5 Tg SO2, as recommended by the HTHH-MOC protocol (Zhu et al. 2025). "

5. The manuscript employs statistical analysis to identify significant responses (e.g., stippling where the forced-minus-control difference exceeds ±2 standard errors of the mean, roughly corresponding to a 95% confidence level). Please state the significance threshold consistently (e.g., forced minus control exceeds ±2 SEM ≈ 95 % confidence). Also, please add a brief rationale for choosing N = 30 and comment on statistical power relative to the weak signals.

As suggested, we have now included the fact that +/2 SEM ≈ 95 % confidence in the captions to the Figure 1: "Stippling denotes regions where the response is statistically significant, here taken as larger than +/- 2 standard errors in the difference in means (≈ 95% confidence level)". We note that in the remaining plot captions, we prefer to state "Stippling indicates statistical significance (defined as in Fig. 1)" in order to avoid repetition, whilst still making it easy to find what the definition and confidence threshold of our statistical significance is.

We have chosen to run 30 ensemble members per experiment (times 4 experiments, each 10-year-long), as it represented the maximum we could do given the available resources. As stated in Section 2.2, the use of 30 members includes a considerable improvement upon the original 10 requested by the HTHH-MOC protocol, which allows us to thoroughly account for and explore the role of interannual variability. In Sections 4 and 5 (Figures 9 and 13) we then explore the detectability of the Hunga signal depending on the number of ensemble members, including the potential spread of inferred responses if smaller ensemble sizes are used.

6. The study outlines a multi-step causal chain: asymmetric volcanic aerosol forcing displaces the ITCZ, initiating La Nina-like cooling; this cooling then reshapes planetary-wave activity and perturbs the polar vortices, ultimately producing regional surface-climate anomalies, including shifts in the NAO. In addition, stratospheric aerosol vs.

water vapor effects have opposing influences on the polar vortex (strengthening vs. weakening), and there are hemispheric differences with ozone-related effects in the south. This multifaceted chain of causality from 2022 Hunga eruption can be challenging to grasp from text alone. While this is not mandatory, a schematic, summarizing the indirect pathways by which the Hunga eruption influences surface climate, would enhance the presentation and could be included in the conclusions for clarity.

Thank you for the suggestion – we agree and will include such schematic in the revised manuscript.

Minor suggestions

Line 110 (Figure 1): the $\Delta$sAOD ($\times 10^3$) appears too large compared with observations (OMPS; SAGE-III, etc); please double-check.

Thank you for spotting this – the label should say "$10^{-3}$", and not "$10^3$", our apologies. We have now corrected this.

Line 14    "The 2022 Hunga eruption erupted …" → "The 2022 Hunga volcano erupted."

Corrected.

Line 45    "Stocker, et al., 2024" → "Stocker et al. (2024)".

This citation is already in parenthesis, hence we think the current formatting is correct here.

Line 55    "Jucker at al." → "Jucker et al."

Corrected.

Lines 80 & 380    "Meineshausen" → "Meinshausen"; "Domaisen" → "Domeisen".

Corrected.

Line 118    Use "damps" instead of "dampens".

We believe that the word 'dampens' is correct here (although we acknowledge that either of these words could be used here).

Line 360    "Quagia, et al. in prep" → "Quaglia et al., in preparation".

The study is now in review for ACP and available as a preprint, hence we now include the full citation (Quaglia et al., 2025).

**Community Comment**

Dear authors, we appreciate the manuscript by Ewa Bednarz and colleagues and we are convinced it will make an interesting contribution to the Hunga-Tonga literature. After having gone through the manuscript in detail, we wish to encourage some additional analysis with respect to Kuchar et al (2025). We would encourage the authors to examine in more detail the atmospheric-only simulations (HUNGA_fix) even though the climatological SSTs may dampen the surface response.

It appears that HUNGA_fix has the potential to represent the top-down mechanism suggested in Kuchar et al (2025), i.e. the influence of lower-mesosphere cooling on the stratospheric polar vortex via decreased temperature gradient. Because of HTHH-MOC (Zhu et al., 2024), there is access to the presented simulations which allowed us to reproduce Fig. 2 from Kuchar et al (2025), see Fig. C1 below. As you can see most of the features such as the mesospheric cooling at lower latitudes (Fig. C1a), which has the potential via the thermal wind relation to weaken the polar-night jet, are represented by the HUNGA_fix simulation. Consequently, we observed an increase in polar temperature progressing downward and the associated cooling aloft (Fig. C1b). However, the statistical significance at higher latitudes is rather limited possibly due to the use of monthly means. Note that Fig. C1 shows the evolution of temperature for the winter 2023/2024 in contrast to 2022/2023 used in SOCOLv4 in Kuchar et al (2025). As discussed therein, the mechanism suggested and simulated by SOCOLv4 above should be valid for the winter of 2023/2024 and the following winters if the lower-mesospheric cooling is persistent and strong enough due to the excess WV. This difference has been attributed to the overly fast transport of water vapour in SOCOLv4 and differences in the experiment protocol.

We thank the authors for their valuable comments. We agree that our manuscript could benefit from more focus on the atmosphere-only simulations, and comparison with the results in Kuchar et al. (2025).

In particular, in Section 4 we now include a clear discussion of the changes in the NH vortex in 2023/2024 winter as pointed out by the authors: "In the atmosphere only simulation, where ocean feedbacks cannot interfere with any top-down response, in the second winter the simulations show statistically significant strengthening of the early winter vortex followed by statistically not significant weakening in late winter. Such response might be indicative of aerosol-induced lower stratospheric warming (Fig. S1b) dominating the response in early winter and upper stratospheric cooling (Fig. S1f) dominating the vortex behavior in late winter, with the latter consistent with the mechanism postulated in Kuchar et al. (2025). However, more idealized studies would be needed to diagnose the details of such modulation."

We also reproduced Fig. 9 to assess the detectability of the NH stratospheric vortex in the early winter (Fig. C2a), late winter (Fig. C2b) and March only (Fig. C2c) in SOCOLv4. Similarly, we cannot detect any statistically significant signal beyond the envelope of potential responses in the early and late winter based on our sample size. However, when we assess the responses only in March 2023 we see an emerging negative response in zonal-mean zonal wind at 65°N and 50 hPa.

We thank you for the analysis. We have now rephrased L. 252-258 to read:

"For instance, Kuchar et al (2025) used a 10-member ensemble of Hunga simulations with the SOCOLv4 model, and reported a late winter weakening of the NH polar vortex from the eruption, with the associated impacts on the high latitude surface climate. Assuming similar magnitude of both the forced response and natural variability in SOCOLv4 and CESM2, our results suggest that 10 members is not enough to infer the footprint of the eruption on the NH winter vortex with any confidence. In CESM2, the variability is particularly large for the late winter vortex response (i.e. even larger than in early winter; compare left and right panels in Fig. 9), consistent with added uncertainty in the vortex final warming date."

This suggests that emerging signals on a monthly basis may be detected even with a limited sample size. Accordingly, we would like to suggest further exploring the detectability of signals for the winter 2023/2024 as done in Figs. 9 and S5.

Figure 5 in our paper, showing the evolution of anomalous zonal wind (a-b) and geopotential height (c-d) in the coupled and atmosphere-only simulations, is already done using monthly data, and the plots show that late-winter responses in winter 2023/2024 are not significant in either HUNGA_cpl or HUNGA_fix simulations even with the use of single months. We note that the early winter response is, however, significant in that winter in HUNGA_fix simulations. As stated above, we now include an explicit discussion of the NH polar vortex anomalies simulated in these atmosphere-only runs in winter 2023/2024.